# Early alpha/beta oscillations reflect the formation of face-related expectations in the brain

Marlen A. Roehe[1,2]*, Daniel S. Kluger[2,3], Svea C. Y. Schroeder[1,2], Lena M. Schliephake[1], Jens Boelte[1,2], Thomas Jacobsen[4], Ricarda I. Schubotz[1,2]

**1** Department of Psychology, University of Münster, Münster, Germany, **2** Otto-Creutzfeldt-Centre for Cognitive and Behavioural Neuroscience, University of Münster, Münster, Germany, **3** Institute for Biomagnetism and Biosignal Analysis, University of Münster, Münster, Germany, **4** Experimental Psychology Unit, Helmut-Schmidt-University/University of the Federal Armed Forces Hamburg, Hamburg, Germany

* marlen.roehe@uni-muenster.de

**Data Availability Statement:** All relevant data are publicly available on the Open Science Framework (https://osf.io/vxqrh/).

## Abstract

Although statistical regularities in the environment often go explicitly unnoticed, traces of implicit learning are evident in our neural activity. Recent perspectives have offered evidence that both pre-stimulus oscillations and peri-stimulus event-related potentials are reliable biomarkers of implicit expectations arising from statistical learning. What remains ambiguous, however, is the origination and development of these implicit expectations. To address this lack of knowledge and determine the temporal constraints of expectation formation, pre-stimulus increases in alpha/beta power were investigated alongside a reduction in the N170 and a suppression in peri-/post-stimulus gamma power. Electroencephalography was acquired from naive participants who engaged in a gender classification task. Participants were uninformed, that eight face images were sorted into four reoccurring pairs which were pseudorandomly hidden amongst randomly occurring face images. We found a reduced N170 for statistically expected images at left parietal and temporo-parietal electrodes. Furthermore, enhanced gamma power following the presentation of random images emphasized the bottom-up processing of these arbitrary occurrences. In contrast, enhanced alpha/beta power was evident pre-stimulus for expected relative to random faces. A particularly interesting finding was the early onset of alpha/beta power enhancement which peaked immediately after the depiction of the predictive face. Hence, our findings propose an approximate timeframe throughout which consistent traces of enhanced alpha/beta power illustrate the early prioritisation of top-down processes to facilitate the development of implicitly cued face-related expectations.

## Introduction

Our environment is of a highly dynamic nature, veiling a cascade of statistical regularities. Explicitly, such regularities often go unnoticed, although traces of implicit learning are evident in the brain's neural activity. These regularities are extracted as sensory input and projected

**Funding:** The author(s) received no specific funding for this work.

**Competing interests:** The authors have declared that no competing interests exist.

'bottom-up' over multiple cortical levels in order to establish associative neural representations reflecting external influences [1]. These internal representations are henceforth frequently updated and revised to optimise their reliability. This accumulation of knowledge regarding statistically predictable external recurrences can then be drawn upon when the external input is less informative or lacks certainty [2]. To reduce this ambiguity, predictions based on prior knowledge regarding an external stimulus are sent 'top-down' along the cortical hierarchy and are compared with equivocal sensory-driven input to draw relevant inferences. According to predictive processing frameworks, if, for instance, the top-down prediction and bottom-up input carry dissimilar information, a mismatch in the form of a prediction error is propagated upwards to update a subsequent higher level. On the contrary, no revision of any given level would be necessary if the bottom-up signal is congruent with the top-down prediction. In the context of predictive processing, this bidirectional interplay between incoming sensory signals and top-down projected predictions is the underlying mechanism assisting perception [3,4].

Emerging principles within this field of research have highlighted several biomarkers which support this bidirectional predictive framework. For instance, the face-sensitive event-related potential (ERP), N170, is a reliable temporal marker which, amongst other factors, reflects the level of predictability of a face-related sensory input. Specifically, selected studies investigating face perception conveyed that the amplitude of the N170 component was significantly diminished for expected compared to unexpected faces [5–7]. In line with predictive processing, this suggests that the sensory-driven information of an expected face is met by fairly accurate top-down prediction. To establish such predictions, the brain draws upon prior information of expected events in preparation for their actual occurrence [8]. Recent studies have shown that pre-activation of sensory information, and subsequent sensory priors, are mediated by low frequency oscillations encompassing alpha and beta frequency ranges [8–12]. These oscillations, primarily alpha, are believed to enhance the signal-to-noise ratio in task-related networks by carefully selecting relevant and simultaneously silencing irrelevant populations of neurons to establish a more focused access to representations of expected stimuli [13]. Due to the early access to relatively precise prior information, less cognitive resources are required to process anticipated perceptual input and in turn visual event-related potentials are modulated [7,9,13]. Additionally, updating and optimising this given neural representation would be unnecessary, hence, the forward projection of prediction errors is downregulated. Bottom-up processing as well as the projection of prediction errors functionally relate to high gamma frequency (60 – 100Hz) synchronisation, which requires a greater energetic cost than lower frequencies [14,15]. Based on these findings, a reduction in gamma power would be presumed to proceed the onset of expected events. In contrast, due to the limited access to pre-activated prior information, more cognitive resources would be allocated to processing unexpected occurrences [10,15,16]. Unexpected events could, therefore, be distinguishable from expected occurrences by enhanced post-stimulus gamma-band activity (GBA), whereas expected images are preceded by an enhancement in pre-stimulus alpha/beta power and a suppression in GBA post-stimulus onset [11]. In turn, whilst less cognitive resources are devoted to processing sensory information of expected targets, subsequently evoking a diminished ERP response, the opposite would be expected for novel or surprising occurrences [7,9].

Although the fundamental principles of adaptive perception have been well established, various aspects relating to the genesis and development of top-down driven expectations remain underexplored. Several studies have investigated the presence of pre-stimulus alpha/beta power as an indicator of expectation [9,12,17], as well as examined the pre-activation, maintenance, and transfer of prior face-related knowledge [8]. Yet, these studies primarily focus on a small fragment of the pre-stimulus timeframe immediately prior to the onset of expected events. Therefore, it remains unclear at which point facilitatory processes aiding the

development of cued face-related expectations commence and how this development evolves over time. The main aim of the present study was, therefore, to locate the point within the pre-stimulus period at which the enhancement in alpha/beta power is initiated for expected relative to random images. Moreover, we meant to investigate whether this enhancement in alpha/beta power either (i) fluctuates, (ii) shows a gradual and steady increase until the expected event occurs, or (iii) shows an accelerated increase just prior to stimulus onset. To our knowledge, the current study, therefore, provides new insight into the evolution of implicitly cued face-related expectations.

Through employing a statistical learning paradigm during a short training session, participants acquired implicit knowledge of the statistical relationships and, hence, predictable nature of certain stimuli. More specifically, the participants completed an explicit gender classification task whilst implicitly learning and predicting the statistically predictable occurrences of certain face images. Participants consequently relied on a previously establish representation of the interrelationships between certain images to form subsequent perceptual expectations. As such, the formation of these expectations was dependent on memory. Foremost, we aimed to replicate findings verifying the presence of implicit expectations. In line with previous studies, we expected a faster and more accurate behavioural response for expected faces alongside an attenuation of the N170 amplitude [1,7]. Furthermore, we assumed that whilst an enhancement in GBA should succeed the depiction of randomly occurring faces [11], the statistically expected images should be met with a prior elevation of alpha/beta activity [8,11,12,17]. A systematic relationship between this increase in alpha/beta activity and the amplitude reduction of the N170 in response to the stimulus would subsequently support the suggestion that the increase in alpha/beta activity reflects predictive processes. Thus, our primary motivation was to examine pre-stimulus alpha/beta power, indicating the development of face-related expectations, in order to determine the initial onset and offset confining the formation process of implicitly cued expectations.

## Materials and methods

### Participants

A total of 33 individuals participated in this study (23 women; 23.1 ± 3.51 years of age [mean ± SD]) after having signed informed consent based on the principles expressed in the declaration of Helsinki. All participants were right-handed as assessed by the Edinburgh Handedness Inventory [18], reported (corrected-to-) normal visual acuity and had no history of neurological and psychiatric disorders. For compensation, participants were either accredited with class credits or reimbursed for their participation (25 Euros). Four additional participants, whose EEG data contained excessive sweat artefact contamination (severe drifts in the signal), were excluded from further analyses. The study was approved by the Ethics Committee of the University of Münster (Department of Psychology).

### Stimulus material

Participants were presented with 25 neutral face images (12 women) chosen from the Radboud Faces Database (RaFD) [19]. Since visual information to process faces is extracted using sequences of eye fixations over mainly eye regions (the mouth region being second), all images were scaled so that those facial features (especially eyes) aligned [20]. This was done to reduce the amount of eye movements.

To generate statistical regularities, eight of these images were sorted into four reoccurring pairs under the following sequential guidelines: i) a male face invariably preceded a female face, ii) a male face invariably preceded another male face, iii) a female face invariably preceded

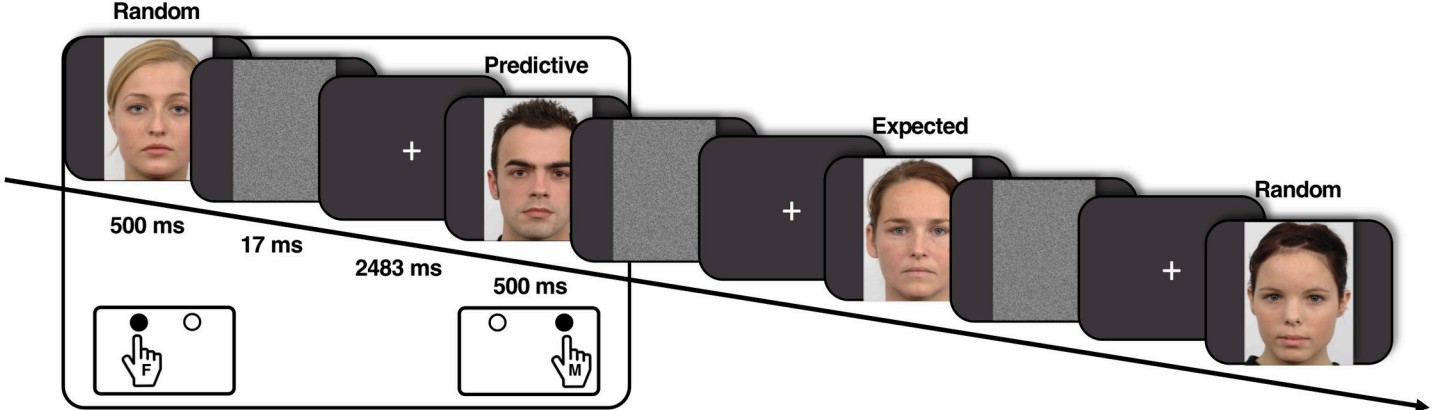

**Fig 1. Schematic illustration of the experimental task.** Each image was depicted for 500ms followed by a white noise mask (17ms) and a darkened fixation screen (2483ms). The participants were instructed to press either the left (right index finger) or right (right middle finger) button on a response box to discriminate between female (F) and male (M) images, respectively.

a male face, and iv) a female face invariably preceded another female face. Each individual participant was assigned a unique set of four pairs which were pseudorandomly embedded amongst reoccurring arbitrary face images. The face images (W = 9.5cm, H = 14cm) were depicted individually in the centre of a black background for 500ms (subtending visual angles of approx. 9˚ vertically and 6˚ horizontally). These were immediately followed by a 17ms white noise mask and a fixation period of 2483ms. Each of these trials was, therefore, a total length of 3000ms (Fig 1).

## Task

The participants engaged in a gender classification task without having prior knowledge of the presence of the embedded pairs. They were given a response box and were instructed to respond as fast and accurately as possible–via a right-hand button press–respective to the gender of the face presented on screen. Here a left button press (right index finger) classified the presented image as a female face, whilst a right button press (right middle finger) classified the depicted face image as male (Fig 1).

## Experimental procedure

Participants were tested on two consecutive days. The first day consisted of a short, 18-minute behavioural training session, providing a chance for the participants to gain implicit knowledge regarding the presence of the paired images and to familiarise themselves with the classification task at hand. Four image pairs were pseudorandomly hidden amongst 17 arbitrary face images, with each image depicted five times to form sequences (blocks) of approximately six to seven minutes (125 images within each block). The first image within each pair served as a predictive event for the second image and subsequently enhanced its predictability. Focusing on this predictability regarding the occurrence of a certain image, all images could be sorted into two image categories–*expected* and *random*. Given the confounding informative nature of a predictive image, these images were removed from analyses determining the predictability of a shown image (behavioural, ERP and gamma-band analyses). The timeframe following the onset of predictive images was, however, examined in the alpha/beta analysis, since it also served as the pre-stimulus timeframe for the expected images.

The EEG session on the following day comprised of an elongated replica of the training session. All images were equally distributed throughout eight blocks with self-determined breaks separating them (yielding a total of 1000 trials). Whilst the four image pairs remained consistent throughout the training and EEG sessions for each individual participant, the combinations of paired images differed and were counterbalanced across participants. The participants were seated comfortably in a dimly lit EEG booth and advised to keep general movement to a minimum. Overall, the EEG session took approximately 50–60 minutes to complete depending on the length of the breaks.

A questionnaire following immediately after the EEG session tested the participants' explicit awareness of the predictive nature underlying the classification task. The participants were asked to state whether they had noticed face images which invariably ensued certain predictive images and asked to identify them seriatim.

The experiment was programmed and performed using Presentation 18.1 (Neurobehavioral Systems, San Francisco, CA, USA).

## Behavioural data analysis

The statistical analysis of the behavioural response time (RT) and accuracy (percentage of correct responses) was performed in R (version 3.6.0; R Foundation for Statistical Computing, Vienna, Austria; Rstudio Team, 2015). Premature and prolonged responses (occurring 3 SDs faster/slower than the aggregated group mean), in addition to incorrect answers, were excluded from the behavioural analysis. Random images were arbitrarily selected to create a sample size equal to the number of expected images (~160 of expected and ~170 of random image trials per participant). Since we hypothesised that the responses for the expected faces would display an increase in accuracy aligned with a decrease in RT, these two aspects of performance were subjected to individual dependent, one-tailed $t$-tests for the two image categories (expected versus random).

## EEG data analyses

**EEG data acquisition.**   Scalp EEG was acquired using 62 Ag/AgCl-electrodes mounted to the actiCAP snap electrode cap in combination with the BrainVision Recorder software (Brain Products, Gilching, Germany). The electrodes were placed according to the 10–20 system and additional electrooculogram (EOG) electrodes were attached below and next to the right eye to account for vertical and horizontal eye movement, respectively. An online bandpass filter (0.1 – 1000Hz) was applied to the EEG data recorded at a sampling rate of 1kHz. Electrodes FCz and FPz served as online reference and ground, respectively, and were disregarded from all analyses. Electrode impedance was maintained below 10 kΩ.

**EEG signal processing.**   EEG data was pre-processed offline using the EEGLAB toolbox (version 14.1.1b) [21] in MATLAB (R2017b). The raw data was down-sampled to 500Hz and bandpass filtered by applying a 0.1Hz high-pass and 30Hz low-pass Butterworth filter (12 db/octave) for the ERP analysis, whereas a 0.5Hz high-pass and 100Hz low-pass Butterworth filter was implemented for the time-frequency analysis (TFA) [22]. Line noise was suppressed at the source through a carefully designed set-up (as recommended by [23]). Continuous data was segmented into epochs extending from -200ms pre- to 600ms post-stimulus onset for ERPs. The 200ms prior to stimulus onset served as a baseline. For the TFA, data was epoched from -2000 to approximately 1500ms, time-locked to stimulus onset. These time segments of 3500ms framing image onset were used with the intention to allow edge artefacts to subside before and after our points of interest [22]. Consecutive epochs overlapped by approximately 500ms to minimise loss of data during convolution. Ocular correction was applied

using the Gratton plug-in for EEGLAB [24]. Noisy channels (kurtosis criterion: z > 6) were manually inspected and replenished by an interpolation of neighbouring electrodes (ERPs: 1.14% and TFA: 0.66% of electrodes were interpolated). For the ERP analysis, semiautomatic artefact inspection discarded epochs contaminated by artefacts exceeding an amplitude threshold of ± 75 µV or voltage fluctuations greater than 50 µV with regard to the previous sample point (3.9% of trials removed). For the TFA, epochs with artefacts exceeding an amplitude threshold of ± 200 µV and voltage fluctuations greater than 50 µV were rejected (6.2% of trials removed). A dataset was disregarded when more than 2 SDs of trials were removed during the semiautomatic rejection (ERP mean: 943 trials; TFA mean: 941 trials). Henceforth, four out of the 37 participants were dismissed from all further processing. During the final pre-processing step, datasets were re-referenced to a common average. For all EEG analyses, the number of expected and random trials was equalised across participants (expected: ~160 per participant; random: ~170 per participant).

**Event-related potentials.**   The epochs framing the event of interest were averaged across each image category (expected and random) for each individual participant. The N170 was quantified by measuring the mean amplitude within the timeframe of 150 – 200ms in relation to a pre-stimulus baseline of 200ms. Based on former literature, we restricted the attributing electrode sites to exclusively P7/P8 and TP7/TP8 for our analysis [25–29]. In line with our directional hypothesis, the mean amplitudes for expected and random images were subjected to dependent, one-tailed *t*-tests for each set of electrodes (left hemisphere: TP7/P7; right hemisphere: TP8/P8; see S1 File for an alternative repeated measures cluster permutation test approach).

**Time-frequency.**   The spectral analysis was performed using the MATLAB toolbox FieldTrip [30]. Spectral power was estimated by applying FFT to a sliding window passing through averaged trials (for both low and high frequencies). A Hanning taper was used for low frequencies (2 – 30Hz) by centring a 500ms fixed sliding window that moved in time steps of 50ms and 1Hz increments. This process subsequently constructed trial windows extending from -1750 to 1200ms (stimulus locked), as 250ms on either side of the original epoch frames (-2000 –approx. 1500ms stimulus locked) were discarded due to convolution. For high frequencies (40 – 100Hz), an adaptive DPSS (discrete prolate spheroidal sequences) multitaper approach was applied [31]. Estimates were acquired using a 500ms fixed sliding window maintaining identical stepwise motion over time and frequency axes as previously stated (± 4Hz smoothing).

Cluster-based permutation tests were computed in three dimensions (frequency, channel, and time) to correct for multiple comparisons. Hence, voxels of the two image categories were subjected to Monte Carlo randomisation tests with 1000 iterations and a significance level of α = .05. With reference to our hypotheses regarding the enhancement of alpha/beta power pre-stimulus and the diminution of gamma power post-stimulus onset for expected versus random faces, dependent, one-tailed *t*-tests were computed for these permutation tests. Statistical tests were performed on the normalised difference in raw power estimates between expected and random images (difference $_{\text{expected vs. random}}$ = (X-Y)/(X+Y)). This normalisation was also applied to all spectral data used for time-frequency representations.

Lastly, data-driven analyses were carried out to assess the relationship between the observed effect in pre-stimulus alpha/beta power and the modulation of the N170. For each participant, the difference in normalised alpha/beta power between expected and random images was calculated and averaged over channels, frequency, and time. In this case, power was only averaged across those channels that contributed to the positive cluster. The difference in mean amplitude for the N170 was computed and averaged over time and left electrodes for each participant. The modulation of alpha/beta power within the immediate pre-stimulus timeframe

(1250 – 3000ms) was z-standardised and correlated with the z-standardised magnitude of the difference in mean amplitude for the N170. To examine the functionality of the last two peaks more closely, their underlying alpha/beta power was segregated and correlated individually with the modulation of the N170. The timeframe proceeding the onset of the predictive images was not included in these analyses because of the images' informative and 'cue-like' nature.

# Results

## Behavioural results

Throughout the EEG session, participants engaged in a classification task with the instructions to identify the gender of the depicted faces as fast and as accurately as possible. The performance for both image categories conveyed the participants' close engagement with the task at hand (expected: 96% accuracy; random: 96% accuracy; one-tailed: $t(32) = 0.51$, $p = .305$). The RT for the two observed image categories showed a mean of 478ms ($SD = 49$ms) for expected faces and 479ms ($SD = 46$ms) for random faces. The dependent, one-tailed $t$-test showed no significant difference between the two categories ($t(32) = 0.38$, $p = .352$).

Notably, the answers of the questionnaires revealed that only a single participant became explicitly aware and was capable of correctly identifying merely one out of the four confronted pairs.

## Event-related potential results

The modulation of the N170 played a fundamental role in identifying whether the participants had gained implicit knowledge regarding the predictability of the paired images. Our approach examined four relevant channels–TP7 and P7 in addition to their right lateral counterparts–for a substantial reduction in mean amplitude (150 – 200ms) for expected faces. Supporting our hypothesis, a significant reduction in mean amplitude was observed for expected (-1.31 ± 2.40 μV) versus random images (-1.51 ± 2.25 μV) at left parietal and temporal-parietal channels (Bonferroni-corrected: $p = .039$; Fig 2). In contrast, no significantly reduced amplitude for expected faces was evident at right-lateralised channels (expected: -1.80 ± 3.02 μV; random: -1.86 ± 3.05 μV; Bonferroni-corrected: $p = .569$).

## Time-frequency results: Gamma oscillations

An underlying implication of predictive processing is the notion that the feeding forward of sensory information is upregulated for novel or unexpected as compared to expected occurrences [14]. In accordance with this conception, our time-frequency analysis showed a significant reduction in gamma power for expected versus random images within the first 1000ms upon stimulus onset ($p = .04$, cluster corrected; Fig 3). In other words, a significant enhancement in GBA was found in response to random compared to expected images. No particular frequencies within the broad gamma-band were singled out a priori, meaning, gamma power (40 – 100Hz) was treated as a singular entity. Interestingly, we observed that the gamma-related spectral difference between random and expected images appeared to reside in an early and late enhancement (Fig 3A). Visual inspection of the cluster revealed that the channels contributing to these elevations in GBA were predominantly located over posterior scalp regions (Fig 3B).

## Time-frequency results: Alpha and beta oscillations

The above analyses determined several neural traces which suggest that the predictive nature underlying the classification task was learned implicitly. Building upon this finding, we

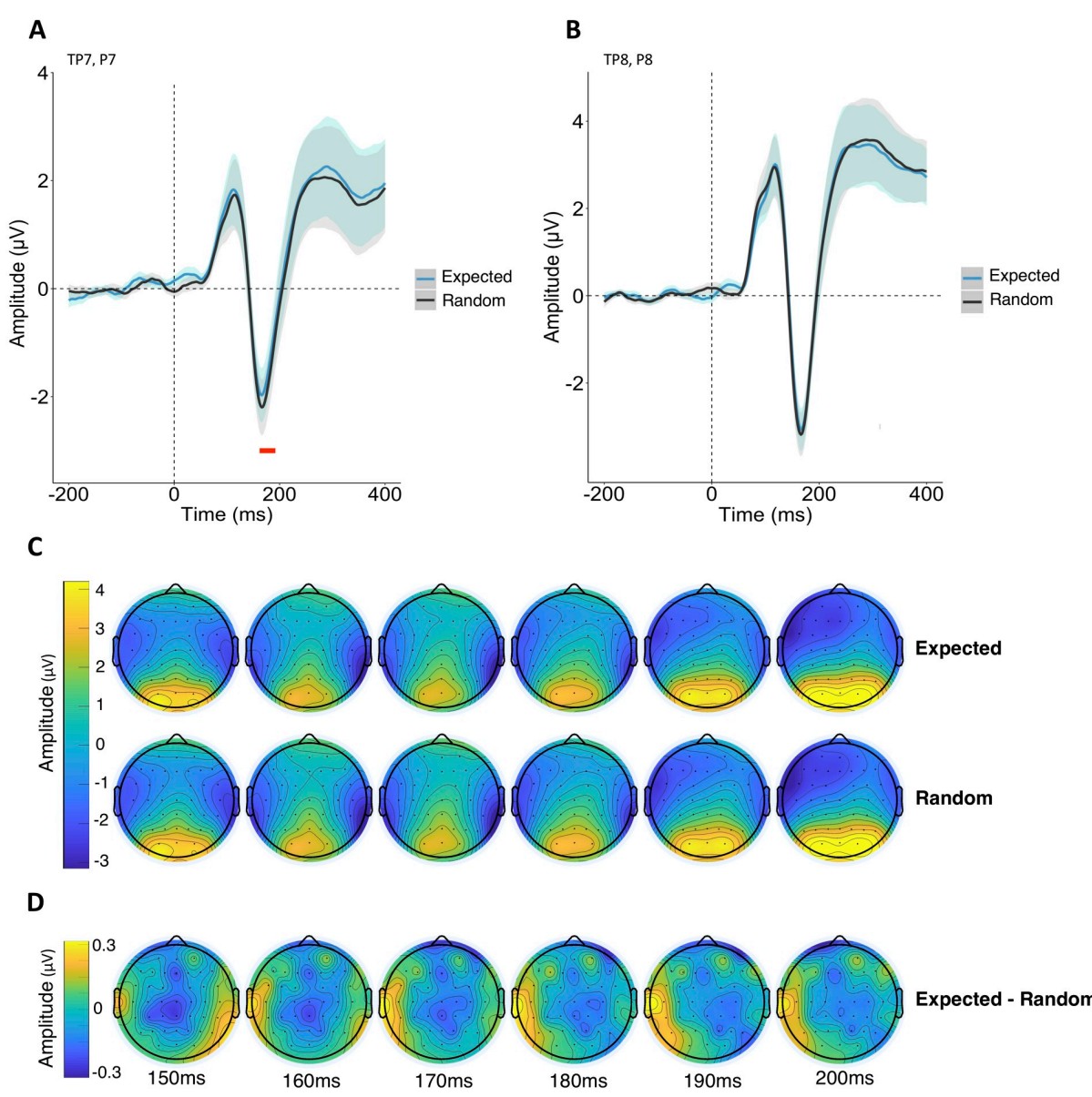

**Fig 2.** (A) A significant reduction in the N170 amplitude for expected (blue line) in comparison to random (black line) faces was found across the averaged electrodes TP7 and P7 (left hemisphere). The timeframe of the significant mean amplitude difference is marked by a red dash. The shaded area illustrates within-participants confidence intervals for the expected (blue) and random (grey) faces. (B) No vast differences in the N170 were evident across electrodes TP8 and P8 (right hemisphere). (C) Voltage topographies (μV) for expected and random images show activity in a timeframe from 150 – 200ms following stimulus onset. (D) Topographies show voltage differences between expected and random face images.

examined alpha and beta frequencies to determine a confined time-window preceding an expected event, in which the development of an expectation was reflected by its power distribution pattern. Hence, this analysis focused on the presence and distribution of low frequencies, primarily associated with top-down processes, in the short timeframe preceding the expected stimuli [12,17,32]. In line with our hypothesis, we found a significant enhancement

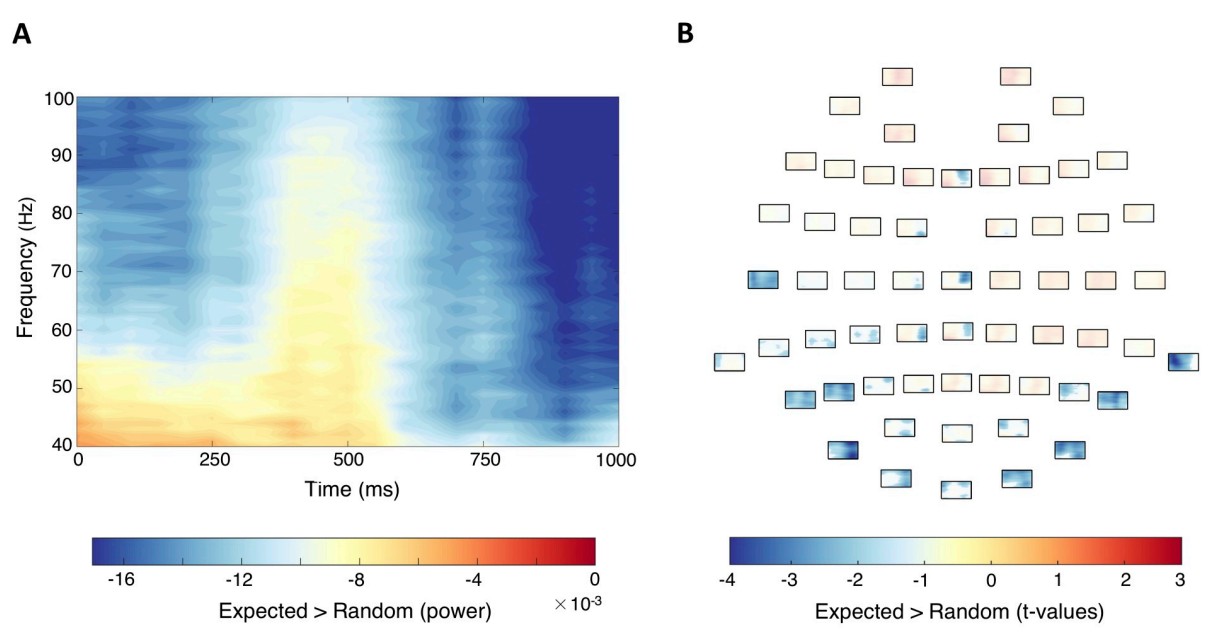

**Fig 3.** (A) Time-frequency representation (TFR) showing the normalised difference in gamma-band activity (GBA) between expected and random images peri- and post-stimulus onset (40 – 100Hz; 0 – 1000ms). Power was averaged across electrodes contributing to the negative cluster (*p* = .04, cluster corrected). (B) TFRs for each individual channel illustrate the topographical distribution of the negative cluster. The significant time and frequency points composing the negative cluster stand out as opaque; insignificant differences are transparent.

in alpha and beta power for expected images (in comparison to random images) ranging from approximately 1250ms after the presentation of the predictive image to onset of the expected stimulus (*p* = .017, cluster corrected; Fig 4B). However, as displayed in Fig 4A, the positive cluster seems to commence prior to this enclosed timeframe. Thus, we additionally contrasted the time course extending from stimulus onset until 1200ms post-stimulus onset between predictive and random images. Interestingly, a further positive cluster enclosing alpha/beta frequency bands was observed in this timeframe (*p* = .044, cluster corrected). Visual inspection of the cluster suggested that the channels corresponding to the largest power differences between predictive and random images were predominantly located over central electrodes within occipital, parietal, and frontal regions (Fig 4Ai and 4Aii). These topographical distributions then spread to primarily occipital and frontal regions for the largest difference in alpha/beta power between expected and random images (Fig 4Bi). In contrast to the previous scalp maps, the final enhancement in alpha/beta power was mostly lateralised bilaterally over parietal regions (Fig 4Bii). Collectively, these findings suggest that the facilitation of the development of a cued expectation is initiated by the onset of the predictive image and ends shortly prior to depiction of the expected image.

*Post hoc* correlations demonstrated that the modulations of alpha/beta power (8-30Hz) within neither the entire pre-stimulus timeframe immediately prior to stimulus onset (1250 – 3000ms) nor the middle peak (1250 – 1750ms) significantly correlated with the magnitude of the reduction of the N170 (Spearman's rho = .19, *p* = .838, 95% CI [-0.16 0.50]; Spearman's rho = .04, *p* = 1, 95% CI [-0.31 0.38], respectively). A significant positive relationship was, however, observed between the modulation of alpha/beta power underlying the final peak (8-30Hz; 2500 – 3000ms) and the modulation of the N170 (Spearman's rho = .46, *p* = .021,

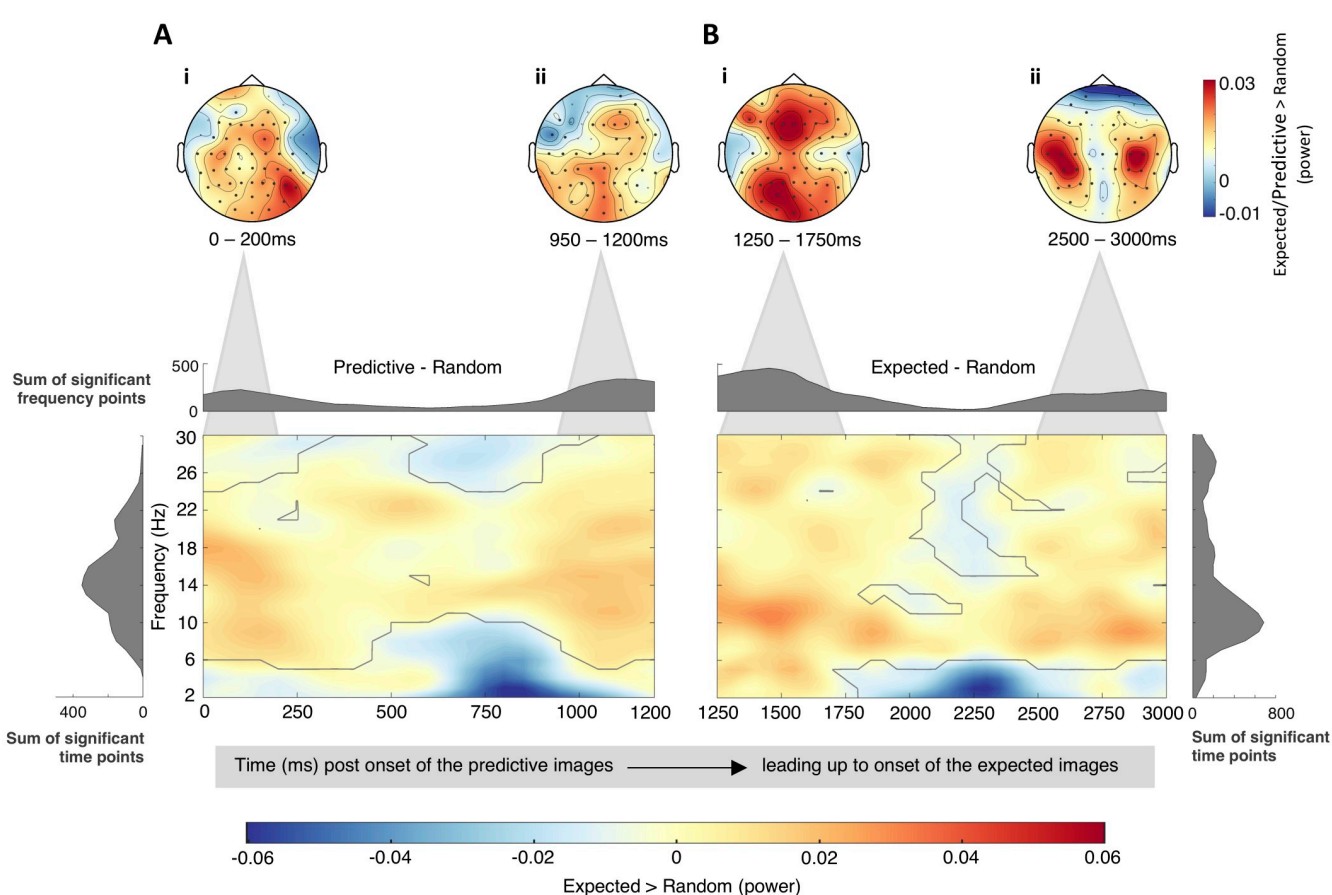

**Fig 4. TFRs of the normalised differences in low frequency power (2 – 30Hz) averaged across channels contributing to the positive clusters (outlined in grey).** The timeframe shown extends from stimulus onset (predictive/random) to stimulus onset (expected/random). The histograms along the x-axes show the sum of significant frequency points per time point across cluster contributing channels. The reversal is shown in the histograms along the y-axes (sum of significant time points for each frequency across cluster contributing channels). (A) Significant cluster for the alpha/beta power differences between predictive minus random images. Scalp maps illustrate the topographical distribution of the greatest power differences (predictive—random) within (i) 0 – 200ms (14 – 19Hz), and (ii) 950 – 1200ms (10 – 18Hz). (B) Significant cluster for the alpha/beta power differences between expected minus random images. The scalp maps illustrate the topographical distribution of power differences (expected—random) within (i)1250 – 1750ms (10 – 14Hz) and (ii) 2500 – 3000ms (6 – 11Hz). A 50ms rift disjoins the 3000ms interstimulus timeframe as a result of the chosen epoch size and Fourier transform parameters (see Materials and methods). Both 4A and 4B, however, provide supportive indications to assume that in place of the 50ms rift, a steady increase in significant frequency points (histograms along the x-axes) would link the gradual increase in 4A with the peak seen in 4B.

95% CI [0.14 0.69]; Fig 5). All above *p*-values were Bonferroni-adjusted to correct for multiple comparisons. Collectively, these findings suggest that the final peak could reflect a relatively precise expectation of the upcoming stimulus. The continuous enhancement of alpha/beta power extending throughout the entire interstimulus interval may, on the other hand, provide an elongated favourable state optimal for expectation formation.

## Discussion

The present study provides findings which suggest that the development of implicitly cued expectations is optimised by the early prioritisation of top-down processes. In turn, predictable visual events are met by relatively accurate implicit expectations to allow the brain to reserve cognitive resources. These processes were reflected by enhancements in pre-stimulus alpha/

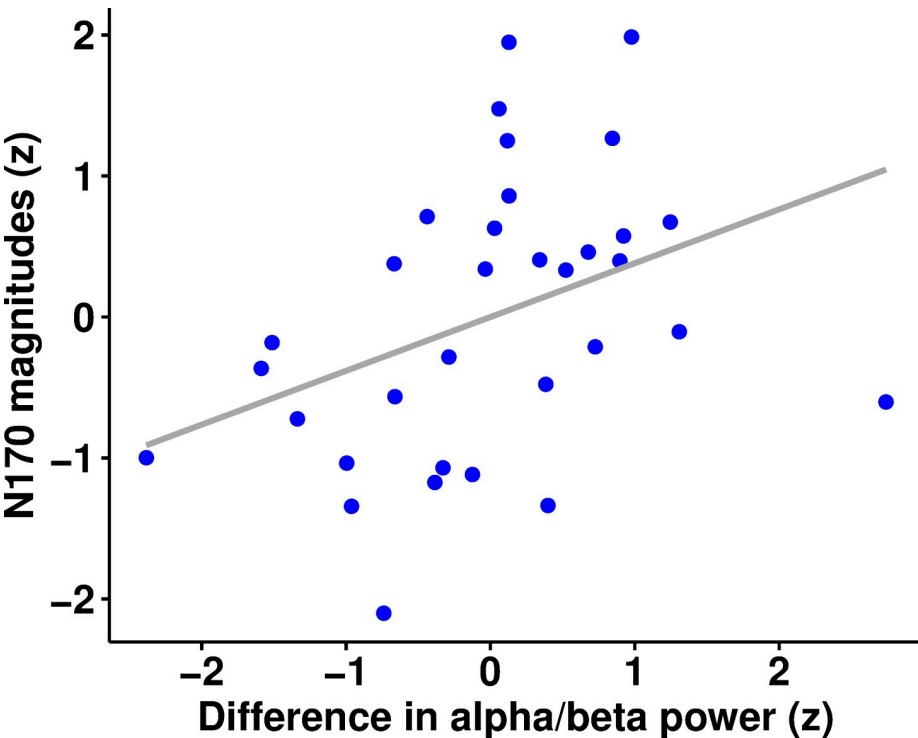

**Fig 5. Correlation between the z-standardised modulations of pre-stimulus alpha/beta power (8–30 Hz; 2500 – 3000ms) and of the left-lateralised N170 (N = 33).**

beta power for expected relative to randomly occurring faces. Intriguingly, this enhancement commenced as early as the onset of the predictive image and prevailed until the expected stimulus occurred. A correlation between the final elevation in alpha/beta power and the reduction of the N170 revealed a positive relationship between these two modulations. Ultimately, a reduction in bottom-up processing for expected relative to random images appeared to be reflected by a suppression in post-stimulus gamma power (Fig 6).

Through employing a short statistical learning test (training) prior to the EEG session, participants were given the chance to acquire implicit knowledge regarding the predictive relationship between paired face images. At first glance, the gain of neither a significant decrease in response time nor a significant increase in accuracy for statistically expected images seems at odds with previous studies [1,7]. Considering the simplicity and repetitiveness of our task, however, the lack of behavioural effects could be caused by a ceiling effect. Turk-Browne and colleagues [1], for instance, took into account that signs of statistical learning become evident after merely 2 to 3 repetitions. In their experiment, novel images and paired image combinations were introduced in each new block to eliminate the likelihood of reaching a plateau in response time and accuracy across expected and unexpected images. Noting that our focal point of interest lay with the origination of implicit expectations and not the statistical learning process as such, our task (the 25 images and paired-up faces remained the same throughout the experiment) may have permitted participants to quickly reach optimal proficiency. A further point to consider is that the images were presented for an entirety of 500ms which may have buffered a speeded reaction. A reduced presentation period could, thus, help encourage participants to give a more speeded response.

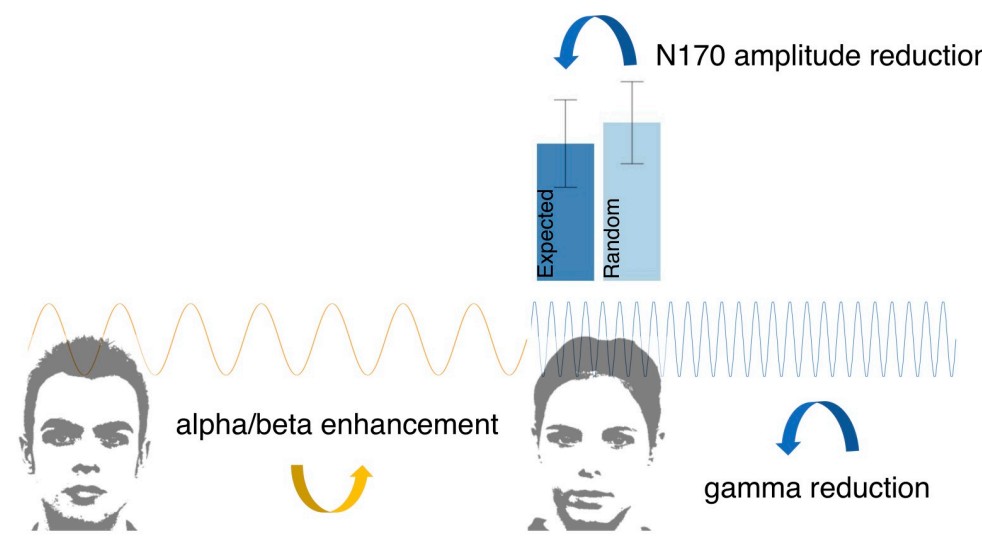

**Fig 6. Schematic overview of observed electrophysiological modulations in response to predictive/expected relative to random images.**

Whilst this study does not allow us to draw a strong conclusion regarding the lateralisation of the N170 effect, the observed modulation of the N170 does, however, appear to support the notion that participants implicitly differentiated between statistically expected and random images (Fig 2). This observation is in line with the established premise that a smaller fraction of cognitive resources is devoted to expected in comparison to somewhat unexpected or surprising events [5,7]. This in turn results in an attenuated electrophysiological response. An assumption as to why this attenuation was solely observed over left-lateralised electrodes is based on the principles that the right-lateralised N170 seems to be more sensitive to familiarity than the left-lateralised N170. That is, studies regarding the role of the N170 during face identity processing have shown that consecutive presentation of identical face images leads to a reduction in the N170 [33,34]. On the contrary, Jemel, Schuller & Goffaux [35] observed an enhanced N170 amplitude for familiar (famous) in comparison to unfamiliar faces during overt face recognition. These habituation effects or memory-driven modulations were found to be predominantly right-lateralised, irrespective of gender [33–35]. Although consecutive depictions of the same face were not permitted within our pseudorandomisation parameters, each individual image was presented 55 times (regardless of its assigned category) throughout the two experimental sessions (training and EEG). Thus, it seems plausible that the N170 components for expected and random images may have, to some extent, been influenced by habituation. This right-lateralised habituation effect may, therefore, have dampened a considerable right-lateralised, expectation-related modulation.

On a different note, past studies have shown sex-related differences in face processing and the lateralisation of the N170. These findings suggest a dominating right lateralisation of the N170 in men and a more bilateral tendency in women [36,37]. Intriguingly, Proverbio and colleagues (2012) showed that sex-coding studies revealed a slightly different pattern in hemispheric lateralisation [38]. Here, women showed a more dominating left-lateralised response whilst men showed bilateral functioning; thus, suggesting that the involvement of the left hemisphere is essential during gender classification in both gender groups. Given that in the present study participants performed a gender classification task, the observed left-lateralised

modulation of the N170 may have been influenced by the underlying nature of the task at hand and the fact that women outnumbered men (10 men and 23 women). However, this unbalanced sample makes it difficult to draw firm conclusions regarding any sex-related differences impacting hemispheric lateralisation. Ultimately, this question would be interesting to pursue in future, with an adequately designed study that specifically investigates how sex-related difference may impact the origination of face-related expectations.

On a final note, the current study used faces as stimuli because the N170 component is a well-established signature of face processing. We would, however, like to emphasize that the N170 has also been reported for non-face stimuli [5,39]. Whether the reduction of the N170 along with the modulations in alpha/beta power observed here for expected faces generalises across other stimulus categories remains to be investigated.

Acknowledging that the occurrence of the face images without a preceding predictive image lacked the predictability of the paired images, random images were deemed to require more cognitive resources and elicit an enhanced gamma-band response. In other words, since all images were task relevant and required a specific behavioural response, it seems likely that more cognitive resources were necessary for processing randomly occurring images, for which the gender was not foretold by a predictive image. In line with previous findings, we observed enhancements in GBA for the somewhat unexpected random images within peri- and post-stimulus periods (Fig 3A; for a review see [40]). Drawing on previous studies, gamma synchronisation has been shown to play a facilitatory role during specific neural functions such as feature binding of incoming visual information [16], the projection of prediction errors [10,14], and influencing synaptic strength during memory encoding and retention [15,41]. When linking these previous findings to our observations, the early peri-stimulus enhancement in GBA (~ 0 – 500ms) could reflect the feeding forward of salient visual information which unifies each individual random image. This notion is supported by the observation that this early gamma-band enhancement is predominantly distributed over occipital electrodes (Fig 3B). Namely, regions which are associated with low-level perceptual processing. Initial processing of incoming visual information, therefore, seems to be augmented for the somewhat unexpected in comparison to expected stimuli.

Given that fast gamma frequencies (~ 60 – 100Hz) are deemed optimal for strengthening synapses during the encoding and updating of short-term memories [14], the later post-stimulus enhancement in broadband gamma power (~ 500 – 1000ms) could indicate that neural representations of the random images are encoded, retained, and revised [16]. Since the random images were recurrently presented over the duration of the experiment, it seems plausible that associated representations could be kept "active" whilst being progressively updated by bottom-up sensory input upon depiction. Consistent with previous findings by Arnal, Wyart & Giraud [10] and Bauer and colleagues [11], the late enhancement in broadband GBA could, thus, reflect the augmented projection of prediction errors from early visual areas via low GBA and the revision of higher cortical levels via high GBA for random images (Fig 3B).

Complementary to previous studies, we found that the depiction of expected targets (in comparison to random faces) was met by an enhanced pre-stimulus alpha- and beta-band activity [11,12,17]. Extending previous findings, we observed that this enhanced alpha/beta activity persisted throughout the entire interstimulus interval. Interestingly, this elongated enhancement in alpha/beta power was governed by three peaks that marked the largest differences in power between expected and random images (Fig 4). The first peak, cresting shortly after stimulus onset (~ 0 – 200ms), suggests an elevation in alpha/beta activity for predictive relative to random images (Fig 4A). It appears that the initial activation of underlying processes facilitating expectation formation is subsequently triggered by the informative attribute of these cue-like images. Namely, the predictive image itself marks a pivotal juncture and

foretells the approach of a certain expected face. The largest power difference between predictive/expected and random images appears to be primarily located across central electrodes within occipital, parietal, and frontal regions (Fig 4Aii and 4Bi). Even though the corresponding scalp map does not provide the same spatial resolution as magnetoencephalography results, this topographical distribution seems to show the engagement of predominantly dorsal regions, frequently associated with the propagation of top-down processes [10,11]. This continuous modulation in alpha/beta power, thus, seems to suggest that the prioritisation of top-down processes commences much earlier than just immediately prior to the occurrence of the expected target. Several past accounts have provided evidence to suggest that alpha/beta power is an electrophysiological marker for the inhibition of forward feeding networks [13,42]. Arguably, it seems very plausible that a similar neural state is elicited upon the presentation of the predictive image. As such, the predictive image seems to give rise to a favourable condition in which increases in alpha/beta power reflect prioritisation of top-down processes whilst competing forward-feeding representations are suppressed. Especially since each predictive image only cued a single specific face, alternative neural representations were unnecessary to be processed or maintained during this interval. The reverse has been demonstrated recently in a study by Griffith et al., (2019), which showed that a decrease in alpha/beta power (disinhibition of relevant networks) facilitates information processing [42]. Thus, the continuous maintenance of a favourable condition within the timeframe confined by the onsets of the predictive and expected images could appear to aid the development of precise perceptual expectations.

In the context of predictive processing, alpha oscillations are leading modulators of attention and expectation. Yet, the process of how these two means modulate information processing remains controversial. Recent studies have suggested that whilst attention boosts the precision of prediction error by synaptic gain, expectation regulates the precision of top-down predictions [43,44]. In the latter case, a highly predictable event would, thus, yield fewer prediction errors which would be distinguishable by a subsequent attenuation in high frequency neural responses. Given that our predictive images invariably prompted certain face images, the expectation generated should ideally have been fairly accurate. Upon stimulus depiction, less iterative optimisations between hierarchical levels should, therefore, have been necessary to establish a relatively precise representation of the expected stimulus. The observed positive relationship between the final enhancement in alpha/beta activity (2500-3000ms; Fig 4B) and the left-lateralised reduction of the N170 appears to be coherent with this modulation framework. Namely, data-driven observation suggests that there is a systematic relationship between the modulation in alpha/beta power occurring immediately prior to stimulus onset and the modulation of the peri-stimulus N170. This electrophysiological pattern is also in line with the notion that preactivated prior knowledge and subsequent predictions regarding an approaching target must be maintained until this predictable event is encountered [8]. Hence, this *post hoc* observation supports the hypothesis that processes of expectation are reflected in increased alpha/beta activity, which makes processing of expected stimuli more efficient and consequently reduces the amplitude of the N170. In this case, the prioritising of top-down predictions could, therefore, be elucidated in terms of prediction accuracy [11,43].

To summarise, we extended findings substantiating the presence of perceptual expectations. Of particular interest was the observation that the predictive images seemed to be essential for the initiation of the evolution of cued expectations. This was rendered by the enhanced alpha/beta activity cresting shortly after the depiction of the predictive faces. Even though the early peaks did not correlate with the modulation of the N170, the onset of the predictive image seems to initiate an early optimisation of a favourable neural state to boost the development of relatively precise perceptual expectations. The perception of predictable faces is subsequently facilitated through the implementation of these expectations, leading to a suppression in

bottom-up information processing reflected by a reduction of the N170 and GBA. The facilitation of the development of implicitly cued face-related expectations, thus, appeared to prevail over the entire interstimulus period with fluctuations in alpha and beta power varying throughout the three second timeframe. One could question why this pre-stimulus enhancement in alpha/beta power fluctuates instead of being a stable and continual increase in power leading right up to the presentation of expected targets. An explanation could be that the spectral distribution pattern within this timeframe is biased by the temporal aspects of the experimental parameters. Given that three seconds are a relatively long interstimulus interval for this particular perceptual task, the gradual decrease in alpha/beta power (approx. 1500 – 2250ms; Fig 4B) could illustrate a progressive conservation of top-down processes before the final power enhancement which marks the imminent approach of the expected target. This would also explain why only the last peak in alpha/beta power positively correlated with the modulation of the N170. Averaging over the entire immediate pre-stimulus timeframe may have concealed a systematic relationship between the aggregated pre-stimulus alpha/beta power and the modulation of the ERP. The precise functional purpose of the second and most prominent peak would, however, benefit from further investigation, which we intend to do in a currently orchestrated study. In addition, although phase analysis is beyond the scope of the present study, it would be an intriguing research question for future studies to investigate alpha/beta phase coherence at the timepoints of each of the observed peaks.

In conclusion, the current study provides new insight into the temporal dynamics and development of face-related expectations. Notably, our findings raise the notion that the formation of cued expectations does not occur at random within the period preceding a statistically expected target. Instead, the facilitation of this developmental process appears to be instigated by the predictive image and proceeds, with fluctuations in growth, until shortly before the depiction of the target. In turn, expected stimuli are met by a relatively precise expectation to allow the brain to reserve cognitive resources. The evolution of implicit face-related expectations, thus, seems to prevail over the entire interstimulus period. From these results we could draw a timeframe confining the genesis and reflecting the developmental nature of cued face-related expectations. As such, these results open up opportunities for future studies to investigate and pinpoint more specific aspects underlying the anticipation of faces. It would, for instance, be of interest to narrow down the precise functional roles–as well as the neural networks–of the observed pre-stimulus peaks in alpha/beta power. Collectively, this would further advance our understanding of how the development of perceptual expectations is shaped in preparation for upcoming expected targets.

## Supporting information

**S1 File. Alternative ERP analysis.** Repeated measures cluster permutation test approach. (PDF)

## Acknowledgments

We would like to thank Monika Mertens, Marie Kleinbielen, Katherina Thiel, Annika Garlichs, Alina Dette, Corinna Gietmann and Falko Mecklenbrauck for helping to recruit participants and/or collect data. In addition, we are exceedingly grateful for Dr. Axel Kohler's insightful ideas throughout the early stages of this study. Last but not least, for always being willing to engage in constructive discussions, we would like to thank Prof. Dr. Pienie Zwitserlood and all the members of the Schubotz Lab.

## Author Contributions

**Conceptualization:** Marlen A. Roehe, Daniel S. Kluger, Lena M. Schliephake, Ricarda I. Schubotz.

**Data curation:** Marlen A. Roehe, Daniel S. Kluger.

**Formal analysis:** Marlen A. Roehe, Daniel S. Kluger, Svea C. Y. Schroeder, Jens Boelte, Thomas Jacobsen.

**Funding acquisition:** Ricarda I. Schubotz.

**Investigation:** Marlen A. Roehe, Lena M. Schliephake.

**Methodology:** Marlen A. Roehe, Daniel S. Kluger, Svea C. Y. Schroeder, Thomas Jacobsen, Ricarda I. Schubotz.

**Project administration:** Ricarda I. Schubotz.

**Resources:** Ricarda I. Schubotz.

**Supervision:** Ricarda I. Schubotz.

**Visualization:** Marlen A. Roehe.

**Writing – original draft:** Marlen A. Roehe.

**Writing – review & editing:** Marlen A. Roehe, Daniel S. Kluger, Svea C. Y. Schroeder, Lena M. Schliephake, Jens Boelte, Thomas Jacobsen, Ricarda I. Schubotz.

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
