## [Decision Letter · Decision Letter 0]

22 May 2021

PONE-D-21-12218

Early alpha/beta oscillations reflect the gradual formation of face-related expectations in the brain

PLOS ONE

Dear Dr. Roehe,

Thank you for submitting your manuscript to PLOS ONE. After careful consideration, we feel that it has merit but does not fully meet PLOS ONE’s publication criteria as it currently stands. Therefore, we invite you to submit a revised version of the manuscript that addresses the points raised during the review process.

We look forward to receiving your revised manuscript.

Kind regards,

Mukesh Dhamala, Ph. D.

Academic Editor

PLOS ONE

Additional Editor Comments:

The manuscript was reviewed by two reviewers. While the reviewers were positive about the overall study, there are several major concerns along with some minor issues raised by the reviewers. The authors need to address all those comments and revise the manuscript before it can be accepted for publication.

Journal Requirements:

2. Please change "female” or "male" to "woman” or "man" as appropriate, when used as a noun (see for instance https://apastyle.apa.org/style-grammar-guidelines/bias-free-language/gender).

3. In your Data Availability statement, you have not specified where the minimal data set underlying the results described in your manuscript can be found. (OSF link is not publicly available.)

PLOS defines a study's minimal data set as the underlying data used to reach the conclusions drawn in the manuscript and any additional data required to replicate the reported study findings in their entirety. All PLOS journals require that the minimal data set be made fully available. For more information about our data policy, please see http://journals.plos.org/plosone/s/data-availability.

4. We note that Figure 1 includes an image of participants. 

Reviewers' comments:

Reviewer's Responses to Questions

**Comments to the Author**

1. Is the manuscript technically sound, and do the data support the conclusions?

Reviewer #1: Partly

Reviewer #2: Yes

2. Has the statistical analysis been performed appropriately and rigorously? 

Reviewer #1: Yes

Reviewer #2: Yes

3. Have the authors made all data underlying the findings in their manuscript fully available?

Reviewer #1: No

Reviewer #2: No

4. Is the manuscript presented in an intelligible fashion and written in standard English?

Reviewer #1: Yes

Reviewer #2: Yes

5. Review Comments to the Author

Reviewer #1: The authors have tried to explore the neural correlates (ERPs and EEG power) underlying face related expectations by presenting participants with two image categories- expected and random; predicted images are followed by the expected images about which the participants learn apriori in the training session. Even though the authors do not find much difference in their behaviour in response to the two image categories, they do find some interesting modulations in N170, alpha/beta and gamma band activity during and prior to the presentation of the expected images compared to random images. Overall the study is well conducted but has some major concerns which are listed below as comments.

Major comments

1. This is a gender discrimination study, yet the participant pool has a gender bias, 23 females and 10 males. What is the rationale for this selection? Out of excluded if there are any males, then situation is even more biased!

2. It has not been mentioned in the methods how the line noise was removed from the EEG data.

Also, for EEG recordings the online reference and ground electrodes were FCz and FPz, respectively. How were the reference and ground chosen? If they are the system defaults, how were the data from these electrodes interpreted?

3. The results of the article show enhancement in alpha/beta power with face related expectations. This is particularly shown by a positive correlation (Fig.5) between peak alpha/beta power (2500-3000 ms) and left-lateralized modulation of N170.

It has been shown by several researchers that N170 is just not face-specific but is rather elicited in response to prediction errors. Studies have shown that N170 was larger in amplitude for unpredictable versus predictable stimulus onsets, irrespective of the object category. Please see to the following articles

P. Johnston, J. Robinson, A. Kokkinakis, S. Ridgeway, M. Simpson, S. Johnson, J. Kaufman, A.W. Young

Temporal and spatial localization of prediction-error signals in the visual brain

Biol. Psychol., 125 (2017), pp. 45-57

Thierry, G., Martin, C. D., Downing, P., and Pegna, A. J. (2007). Controlling for interstimulus perceptual variance abolishes N170 face selectivity. Nat. Neurosci. 10, 505–511. doi: 10.1038/nn1864

Can the authors say these effects would not exist for expectations related to non-face stimuli?

4. The N170 modulation was mostly left lateralized seen specifically at P7/TP7 whereas the alpha/beta enhancement was bilateral. A correlation between the two supports the hypothesis that processes of expectation are reflected in increased alpha/beta activity, which makes the processing of expected stimuli more efficient and consequently reduces the amplitude of N170. What would the authors comment on the topographical inconsistency of these two measures in terms of brain regions processing the expectedness of faces?

5. In the TFA specially for studying GBA, the low pass filter has been set to 1000 Hz. However in humans, high gamma only upto ~90 Hz can be recorded through EEG beyond which the high frequency oscillations (which cannot be recorded through EEG as they are mostly of sub-cortical origin) are contaminated with noise which should be filtered out for any time-frequency analysis.

6. The authors claim that their findings propose an approximate timeframe throughout which consistent traces of enhanced alpha/beta power illustrate the development of face-related expectations which peaks at around 2500-3000ms after the onset of the predictive image. This can be due to the expectedness of the expected face or very well be due to just any other stimulus following the predictive face/image (in case some other stimulus instead of the expected image was randomly paired with the predictive image) because the inter-stimulus duration was fixed (~2500ms) which after repeated trials raises expectation for the subsequent stimuli after fixed intervals. How do the authors delineate these two possibilities?

7. If authors can make a figure to connect the brain mechanisms (N170, alpha/beta, gamma) and predictive processes – somewhat of a summary, that would immensely help following the Discussion.

8. No data or codes link was submitted for review, which seems to be PLOS one policy

Minor points

1. Subplots of Fig 3B are incomprehensible.

2. Either put figures with captions in the text together, or keep them in the end with references. The fig caption in text and, figures in the end is very difficult to read.

Reviewer #2: This is a particularly well designed and interesting study by Roche and colleagues to understand the origination and development implicit expectation using EEG. Recent evidence further suggests that both pre-stimulus neuronal oscillations and peri-stimulus event-related potentials are reliable biomarkers of implicit expectations arising from statistical learning paradigms. Employing such a paradigm in gender-classification task based on face stimuli they investigated attenuation of ERP signals, Pre-stimulus Alpha/Beta oscillations and Post-stimulus GBA determine the temporal constraints of face-related expectation formation. While I found the research questions are all relevant and methods employed here are technically accurate and the overall narrative also presented well, I do have the following questions, suggestions and comments.

Introduction

# In my opinion, a more challenging issue I find for all these perceptual paradigms is how much perception of familiar or surprising occurrences are devoid of memory processing. There has to some involvement of component of memory specifically when the participants get trained in what kind of occurrences are expected versus not. Due to this training, they form a representation in memory of those perceptual prototype. Hence, I wonder how much of the face related expectation is aided by the service of memory and if you could dissociate what neurophysiological correlates are driven particularly by perceptual prediction and expectation versus memory component. Indeed you write about habituation and memory effects and repetition suppression effects in the discussion section.

# One of the major issues of this work that I find is I am not too sure if I completely understand how the broader question asked and the overarching goal and hypothesis of this study design is any different from what has been investigated in some of the earlier works to address the following,

How low-frequency neuronal oscillations act as carriers of sensory evidence and top-down predictions, respectively (von Stein and Sarnthein 2000; Bastos et al. 2012). In other words, whether slow pre stimulus alpha oscillations in task-relevant brain regions are stronger in the presence of predictions, whether they influence early categorization processes, and whether this interplay indeed boosts perception in general.

https://pubmed.ncbi.nlm.nih.gov/26142463/

It would be crucial to delineate what separates out this study from the previous ones to be able to appreciate more the importance and exclusiveness of the findings.

# In lines 76-78, authors write that “The insight that the projection of prediction errors is mediated by high gamma frequency (60 – 100Hz) synchronisation, which requires a greater energetic cost than lower frequencies [8,9]”, This statement is understandable but how does that shines light on the underlying processes shaping variability of ERPs is not clear. Perhaps, you could consider revising this statement.

# Also, please explain why lesser Cognitive resource would suggest diminished ERPs. Is there already worked out inverse relationship between no. of Cognitive Resource and amplitude/power of ERP response or vice versa. Could please cite appropriate reference to justify this relationship?

# The opposite such as amplified ERPs are evident for arbitrary or surprising occurrences. This is of course similar to what you expect for familiar versus unfamiliar stimuli or novel stimuli which is not consolidated yet in the memory. However, those which are already consolidated via prior experience/learning are already very familiar (e.g., expected faces) and should give similar to repetition suppression effect (repeated exposure). similarly, for surprising faces one should expect very similar effect as oddball stimuli. I would appreciate if you could you provide some of your reasoning against this comment?

# Authors write “Drawing upon this, it seems apparent that somewhat unexpected events are predominantly distinguishable from expected occurrences by enhanced gamma-band activity (GBA).” There could be multiple interpretation of enhanced gamma-band activity. One specific reason you mention is supported by few recent studies which suggest that enhanced GBA could be involved in encoding and updating internal representation. However, thinking loud along the line of reasoning you have outlined in the Introduction I wonder whether this also has anything to do with deployment of larger number of cognitive resources while processing somewhat unexpected events. If you could kindly comment on this.

Materials and methods

# You do not have a gender balanced sample for this study. I am wondering whether that has any impact on the gradual built up of implicit expectation and top-down prediction of expected face occurrences. Previous studies have shown that in the primary visual cortex while decoding faces and expressions, women showed a more bilateral functioning than men and there are some gender differences while processing neutral and affective faces. There are also difference that are found in the EEG N1 response as well as during oddball N170 amplitude response. Could you please provide your reasoning?

# The authors hypotheses regarding the enhancement Alpha/Beta power pre-stimulus as top down prediction of expected faces. I am wondering why hypothesize about amplitude/power and why not phase of pre stimulus Alpha/Beta. I would imagine phase of pre stimulus Alpha may also play a crucial role. Also, this change in pre stimulus Alpha power could just reflect the ongoing background excitability which has a biophysical basis and has got nothing to do with any prediction about the faces. Please comment on this.

# In lines 214-215, authors have mentioned about the selection of pre-post durations, however, not provided any details with regards to what is the basis of this selection.

The authors write, “-200ms pre- to 600ms post-stimulus onset for ERPs. The 200ms prior to stimulus onset served as a baseline. For the TFA, data was epoched from -2000ms to approx. 1500ms, time-locked to stimulus onset.”

Any comment on what exactly guided your selection of a specific pre stimulus duration as we know from previous studies that the choice of pre stimulus duration can have significant impact on the outcome of power changes and involvement of specific frequency band of interest. Hence, what frequency bands subserve the prediction process is very much sensitive to the choice of those parameters e.g., pre stimulus duration.

Results

# Kindly explain why a title as “Early alpha/beta oscillations reflect the gradual formation of face-related expectations in the brain” deemed appropriate.

Is it really gradual over the entire period of pre stimulus duration or is it an accelerated process just before the onset of the expected image? It seems to be the latter based on the following results unless I did not follow these results correctly.

A significant positive relationship was, however, observed between the modulation of alpha/beta power underlying the final peak (8-30Hz; 2500 – 3000ms) and the modulation of the N170 (Spearman’s rho = .46, p = .021, 95% 381 CI [0.14 0.69]; Fig 5).

Discussion

I enjoyed reading the discussion section. However, I could not resist myself repeating my last comment again. Discussion section actually begins with what was my last comment for the previous section. It seems to me a more appropriate title hiding in the statement below,

“Predictable visual events are often met by implicit expectations to allow the brain to reserve cognitive resources.” Authors should certainly clarify which one of the two is more appropriate interpretation here the gradual development and built up of predictions prior to expected stimulus onset or an accelerated process showing tendency towards conserving cognitive resources in the process of predictions of expected outcome.

# In lines 426-427 I suggest please provide some relevant references when you are discussing right lateralized habituation effects.

“These habituation effects or memory-driven modulations were found to be predominantly right-lateralised.” (reference please)

Is this memory-driven modulations of N170 ERPs predominantly right-lateralised irrespective of gender in face related expectation. Related to my earlier question about selected sampling.

# I think the authors cited and covered references which are most relevant to their findings based on ERPs, pre stimulus brain activity, GBA.

However, I would like to point out that the recent studies suggest frequently reported negative relationship between pre-stimulus α/Beta power and stimulus detection or behavioural performance may be explained by changes in detection criterion. I am wondering whether there is something similar is happing here for example Pre stimulus power change in Alpha/Beta is driven by the state of neural excitability (brain states of individual participants), rather than top-down prediction of expected occurrences. Could you please comment on this. Related to my earlier question on background excitability.

# The authors write in line 512 One could question why this pre-stimulus enhancement in alpha/beta power fluctuates instead of being a stable and continual increase in power leading right up to the presentation of expected targets.

I do have the same question and also related to my previous question of gradual bult up of expectation versus final peak increase before the presentation of expected targets. You have already provided an elaborate and solid reasoning in lines 515-521. But still curious whether this transient enhancement of power in the Alpha/Beta band should be considered at all for the entire inter-stimulus duration of 3 seconds as you have mentioned is too long a duration. Somehow, It seems these individual peaks are fundamentally functionally distinct and more meaningful pre stimulus anticipatory power change is occurring over a relatively short-interval of time frames with regards to significant functional attenuation the amplitude of ERPs to the presentation of expected targets. Hence, the clear role of these three peaks at least the first two and in particular, the middle one is not so clear after all. The waxing and waning of pre stimulus Amplitude/power change over certain time intervals and windows could be purely based on the change in background excitability and brain states.

I could also anticipate substantial inter individual and inter-trial differences in the processes of how expectations reflected in increased alpha/beta activity, which makes processing of expected stimuli more efficient for some participants and consequently reduces the amplitude of the N170 but not necessarily with the same efficiency and reduction for others.

6. PLOS authors have the option to publish the peer review history of their article (what does this mean?). If published, this will include your full peer review and any attached files.

Reviewer #1: No

Reviewer #2: No

---

## [Author Response · Author response to Decision Letter 0]

22 Jun 2021

Dear Editor,

We are very much obliged to you for managing the review process of our submission to PLoS One. We are also very grateful to the reviewers for their constructive feedback. Their valuable comments and suggestions have been addressed accordingly and we wish to submit a revised version of the manuscript for further consideration. Changes to the manuscript have been marked and a point-by-point response to the reviewers’ comments can be found below.

Thank you for your time and consideration. We look forward to hearing from you. 

Yours sincerely,

On behalf of the co-authors

Marlen Roehe 

Journal Requirements:

We thank the editor for noticing this inconsistency. The Supporting Information files have been renamed as mentioned in PLOS ONE’s style templates. 

2. Please change "female” or "male" to "woman” or "man" as appropriate, when used as a noun (see for instance

https://apastyle.apa.org/style-grammar-guidelines/bias-free-language/gender).

We thank the editor for drawing our attention to these guidelines. We have changed “female” and “male” to “women” and “men” when used as nouns. Thus, “female” and “male” were only used as adjectives. 

3. In your Data Availability statement, you have not specified where the minimal data set underlying the results described in your manuscript can be found. (OSF link is not publicly available.) PLOS defines a study's minimal data set as the underlying data used to reach the conclusions drawn in the manuscript and any additional data required to replicate the reported study findings in their entirety. All PLOS journals require that the minimal data set be made fully available. For more information about our data policy, please see http://journals.plos.org/plosone/s/data-availability.

We thank the editor for this observation. All relevant data are publicly available on the Open Science Framework (https://osf.io/vxqrh/). 

4. We note that Figure 1 includes an image of participants. 

We thank the editor for this comment. The face images used as stimulus material were obtained from the Radboud Faces Database upon request [RaFD; Langner O., et al. (2010). Cognition & Emotion]. The website states that RaFD faces can be presented as stimulus examples in “strictly scientific publications…e.g., journal articles.” We hope this will suffice. (Please see link: http://www.socsci.ru.nl:8180/RaFD2/RaFD?p=faq)

Response to the Reviewer #1 

The authors have tried to explore the neural correlates (ERPs and EEG power) underlying face related expectations by presenting participants with two image categories- expected and random; predicted images are followed by the expected images about which the participants learn apriori in the training session. Even though the authors do not find much difference in their behaviour in response to the two image categories, they do find some interesting modulations in N170, alpha/beta and gamma band activity during and prior to the presentation of the expected images compared to random images. Overall, the study is well conducted but has some major concerns which are listed below as comments.

We would like to thank the reviewer for their careful assessment of our manuscript and their detailed feedback. Their insightful comments and suggestions have been extremely helpful for improving the quality of the revised manuscript. In addition, we appreciate that the reviewer has found our study well conducted and the obtained results of interest. We have done our best to address each of the raised points and hope our responses are to their satisfaction. 

Major comments

This is a gender discrimination study, yet the participant pool has a gender bias, 23 females and 10 males. What is the rationale for this selection? Out of excluded if there are any males, then situation is even more biased!

We thank the reviewer for raising this point. Unfortunately, there was a lack of male participants who signed-up for this study, leading to an initial sample size of 11 men and 26 women. The datasets of one male and three female participants had to be disregarded due to excessive sweat artefacts. Thus, the final sample included 10 males and 23 females. We would like to gently highlight, that we did not intent to conduct a gender discrimination study. The gender discrimination task used in this study was implemented along the lines of a ‘cover-up task’. As such, its primary purpose was to keep the participants engaged whilst they learned the paired-up faces in an implicit statistical learning experiment (training), and whilst they predicted sequences of paired faces in the EEG session. To avoid causing confusion, we subsequently labelled the task a gender classification rather than a gender discrimination task. 

2. It has not been mentioned in the methods how the line noise was removed from the EEG data.

Also, for EEG recordings the online reference and ground electrodes were FCz and FPz, respectively. How were the reference and ground chosen? If they are the system defaults, how were the data from these electrodes interpreted?

The thank the reviewer for noticing that the removal of line noise was not mentioned in Materials and methods. Instead of suppressing line noise through filtering, we tried to eliminate electrical line noise as much as possible through a carefully designed setup. Power cables were removed from the EEG booth and amplifiers as well as the active electrodes (actiCap snap, Brain Products) were powered by rechargeable battery supplies. In addition, the screen used for presenting stimuli was integrated into the booth’s wall behind a glass shield. The mains supply powering the screen was located outwith the booth’s interior. The recorded data was transmitted to the experimenter’s computer via an optic fibre datalink, with cables passing through the booth’s walls. Considering that 50Hz is part of the broadband gamma activity which we intended to examine, we subsequently tried to suppress the mains interference at the source to such an extent that would make the application of a notch filter unnecessary [as recommended by de Cheveigné A., Nelken I. (2019). Neuron]. A summary of the above explanation has been added to Materials and methods (lines 230-231).

Lines 230-231: “Line noise was suppressed at the source through a carefully designed set-up (as recommended by [23]).”

The electrodes chosen as reference (FCz) and ground (FPz) were the systems’ default (actiCap snap, Brain Products). These two channels were later disregarded from all analyses. We have added a comment regarding the removal of FCz and FPz to the manuscript (lines 223-224) and would like to thank the reviewer for pointing out this observation. 

Lines 223-224: “Electrodes FCz and FPz served as online reference and ground, respectively, and were disregarded from all analyses.”

3. The results of the article show enhancement in alpha/beta power with face related expectations. This is particularly shown by a positive correlation (Fig.5) between peak alpha/beta power (2500-3000 ms) and left-lateralized modulation of N170. It has been shown by several researchers that N170 is just not face-specific but is rather elicited in response to prediction errors. Studies have shown that N170 was larger in amplitude for unpredictable versus predictable stimulus onsets, irrespective of the object category. Please see to the following articles

P. Johnston, J. Robinson, A. Kokkinakis, S. Ridgeway, M. Simpson, S. Johnson, J. Kaufman, A.W. Young Temporal and spatial localization of prediction-error signals in the visual brain

Biol. Psychol., 125 (2017), pp. 45-57

Thierry, G., Martin, C. D., Downing, P., and Pegna, A. J. (2007). Controlling for interstimulus perceptual variance abolishes N170 face selectivity. Nat. Neurosci. 10, 505–511. doi: 10.1038/nn1864

Can the authors say these effects would not exist for expectations related to non-face stimuli?

We agree with the reviewer and the studies mentioned above, that a modulation in the N170 can be influenced by various stimulations other than faces. This is one of the reasons why we used a single image category, with images similar in perceptual properties such as complexity and salience. In general, the observed neurophysiological effects underlying perceptual expectation could have, in theory, occurred equally well in non-face stimuli of equal complexity. Hence, we believe this to be an intriguing research question for future studies and would like to thank the reviewer for this interesting notion. We have mentioned this outlook in the Discussion (lines 477-481). 

Lines 477-481: “On a final note, the current study used faces as stimuli because the N170 component is a well-established signature of face processing. We would, however, like to emphasise that the N170 has also been reported for non-face stimuli [5,39]. Whether the reduction of the N170 along with the modulations in alpha/beta power observed here for expected faces generalises across other stimulus categories remains to be investigated.”

4. The N170 modulation was mostly left lateralized seen specifically at P7/TP7 whereas the alpha/beta enhancement was bilateral. A correlation between the two supports the hypothesis that processes of expectation are reflected in increased alpha/beta activity, which makes the processing of expected stimuli more efficient and consequently reduces the amplitude of N170. What would the authors comment on the topographical inconsistency of these two measures in terms of brain regions processing the expectedness of faces?

We thank the reviewer for this interesting remark. In principle, anticipatory processes are believed to involve the activity of large networks across frontal as well as posterior regions. In our eyes, this does not necessarily imply a bilateral amplitude reduction of N170 components. Namely, the bilateral N170s could be influenced differently by further underlying processes. For instance, previous studies have reported a predominantly right-lateralised modulation in the N170 during memory-related processes such as habitation and repetition suppression [Jemel B., Schuller AM., Goffaux V. (2010). Journal of Congitive Neuroscience; Caharel S., et al. (2009). Neuropsychologia; Campanella S., et al. (2000). Psychophysiology]. Especially since the pre-stimulus enhancement in alpha/beta activity and the peri-stimulus reduction of the N170 are temporally distinct, we cannot disregard that those other underlying processes may also be influencing the right N170 response during the peri-stimulus timeframe. 

5. In the TFA specially for studying GBA, the low pass filter has been set to 1000 Hz. However, in humans, high gamma only upto ~90 Hz can be recorded through EEG beyond which the high frequency oscillations (which cannot be recorded through EEG as they are mostly of sub-cortical origin) are contaminated with noise which should be filtered out for any time-frequency analysis.

The online bandpass filter was set to 0.1 - 1000 Hz because of our sampling rate of 1000 Hz. This choice in sampling rate was motivated by the convenience of the one-to-one conversion between time in milliseconds and time in samples [Cohen MX. (2014). Analyzing Neural Time Series Data (pp. 61-69)]. The data was later down-sampled to 500Hz and an offline bandpass filter of 0.5 – 100Hz was applied to the TF datasets. We hope this clarifies the reviewer’s question regarding our choice in low-pass filter. 

6. The authors claim that their findings propose an approximate timeframe throughout which consistent traces of enhanced alpha/beta power illustrate the development of face-related expectations which peaks at around 2500-3000ms after the onset of the predictive image. This can be due to the expectedness of the expected face or very well be due to just any other stimulus following the predictive face/image (in case some other stimulus instead of the expected image was randomly paired with the predictive image) because the inter-stimulus duration was fixed (~2500ms) which after repeated trials raises expectation for the subsequent stimuli after fixed intervals. How do the authors delineate these two possibilities?

We thank the reviewer for this thought-provoking question. The fixed interstimulus interval of ~2500ms followed each image regardless of its assigned image category (predictive, expected, and random). Therefore, upon learning the timings underlying these image sequences, it is highly likely that the participants were able to correctly anticipate the onset of any upcoming image (predictive, expected, and random) after the offset of the previous image. However, when contrasting the pre-stimulus timeframes of predictive/expected and random images, these temporal predictions would have been evident in both timeframes and subsequently been cancelled out. In this case, the remaining enhancements in alpha/beta power (expected minus random) should be resultant from the expectedness of the implicitly cued expected image rather than the temporal anticipation of the onset alone. 

7. If authors can make a figure to connect the brain mechanisms (N170, alpha/beta, gamma) and predictive processes – somewhat of a summary, that would immensely help following the Discussion.

We very much appreciate the reviewer’s great suggestion to include a figure which connects as well as summarises the results observed in this study. We have added the following figure to the manuscript.

Fig 6. Schematic overview of observed electrophysiological modulations in response to predictive/expected relative to random images. 

8. No data or codes link was submitted for review, which seems to be PLOS one policy

We thank the reviewer for pointing out that we have not provided a link to a public repository. Please find all relevant data via the following OSF link: https://osf.io/vxqrh/. 

Minor points

1. Subplots of Fig 3B are incomprehensible.

We are grateful for the reviewer’s comment that Figure 3B may be incomprehensible. We have slightly altered the phrasing in the image caption to make the description more precise (lines 347-349). 

Lines 347-349: “(B) TFRs for each individual channel illustrate the topological distribution of the negative cluster. The significant time and frequency points composing the negative cluster stand out as opaque; insignificant differences are transparent.”

2. Either put figures with captions in the text together, or keep them in the end with references. The fig caption in text and, figures in the end is very difficult to read.

We apologise to the reviewer for this inconvenience. According to the PLOS ONE formatting guidelines, figure captions were to be inserted into the manuscript text, immediately following the paragraph where the figure was first cited. Figures were required to be uploaded individually and separate from the manuscript. The captions were not to be included as part of the figure files. If something went wrong during the submission process, we sincerely apologise. 

Response to the Reviewer #2

This is a particularly well designed and interesting study by Roehe and colleagues to understand the origination and development implicit expectation using EEG. Recent evidence further suggests that both pre-stimulus neuronal oscillations and peri-stimulus event-related potentials are reliable biomarkers of implicit expectations arising from statistical learning paradigms. Employing such a paradigm in gender-classification task based on face stimuli they investigated attenuation of ERP signals, Pre-stimulus Alpha/Beta oscillations and Post-stimulus GBA determine the temporal constraints of face-related expectation formation. While I found the research questions are all relevant and methods employed here are technically accurate and the overall narrative also presented well, I do have the following questions, suggestions and comments.

We would like to thank the reviewer for their careful and detailed assessment of our manuscript. Their constructive and insightful feedback has been extremely valuable for improving the quality of the revised manuscript. In addition, we are delighted that the reviewer found our study well designed and the research questions relevant and of interest. We have done our best to address each of the points raised and hope our responses are to their satisfaction. 

Introduction

#1 In my opinion, a more challenging issue I find for all these perceptual paradigms is how much perception of familiar or surprising occurrences are devoid of memory processing. There has to some involvement of component of memory specifically when the participants get trained in what kind of occurrences are expected versus not. Due to this training, they form a representation in memory of those perceptual prototype. Hence, I wonder how much of the face related expectation is aided by the service of memory and if you could dissociate what neurophysiological correlates are driven particularly by perceptual prediction and expectation versus memory component. Indeed you write about habituation and memory effects and repetition suppression effects in the discussion section.

We thank the reviewer for this valuable comment. In the current study, learning and forming a representation of the statistical interrelationship between the different image categories (predictive, expected, and random) was a prerequisite which the participants relied upon when forming face-related expectations during the EEG experiment. As the reviewer rightfully pointed out, it is therefore challenging to completely dissociate between purely perceptual predictions and perceptual predictions that rely on a memory component. We have added a comment to the Introduction to clarify that the observed perceptual predictions were reliant on memory (lines 116-118). 

Lines 116-118: “Participants consequently relied on a previously establish representation of the interrelationships between certain images to form subsequent perceptual expectations. As such, the formation of these expectations was dependent on memory.”

#2 One of the major issues of this work that I find is I am not too sure if I completely understand how the broader question asked and the overarching goal and hypothesis of this study design is any different from what has been investigated in some of the earlier works to address the following,

How low-frequency neuronal oscillations act as carriers of sensory evidence and top-down predictions, respectively (von Stein and Sarnthein 2000; Bastos et al. 2012). In other words, whether slow pre stimulus alpha oscillations in task-relevant brain regions are stronger in the presence of predictions, whether they influence early categorization processes, and whether this interplay indeed boosts perception in general.

https://pubmed.ncbi.nlm.nih.gov/26142463/

It would be crucial to delineate what separates out this study from the previous ones to be able to appreciate more the importance and exclusiveness of the findings. 

We are grateful to the reviewer for raising this crucial point. Throughout the manuscript, we have strengthened the emphasis on how the focal aim of the current study differs from previous studies. 

Lines 101-111 (Introduction): “Yet, these studies primarily focus on a small fragment of the pre-stimulus timeframe immediately prior to the onset of expected events. Therefore, it remains unclear at which point facilitatory processes aiding the development of cued face-related expectations commence and how this development evolves over time. The main aim of the present study was, therefore, to locate the point within the pre-stimulus period at which the enhancement in alpha/beta power is initiated for expected relative to random images. Moreover, we meant to investigate whether this enhancement in alpha/beta power either (i) fluctuates, (ii) shows a gradual and steady increase until the expected event occurs, or (iii) shows an accelerated increase just prior to stimulus onset. To our knowledge, the current study therefore provides new insight into the evolution of implicitly cued face-related expectations.”

Lines 515-518 (middle of Discussion): “Extending previous findings, we observed that this enhanced alpha/beta activity persisted throughout the entire interstimulus interval. Interestingly, this elongated enhancement in alpha/beta power was governed by three peaks that marked the largest differences in power between expected and random images (Fig 4).” 

Lines 566-568 (end of Discussion): “To summarise, we extended findings substantiating the presence of perceptual expectations. Of particular interest, was the observation that the predictive images seemed to be essential for the initiation of the evolution of expectations.”

Lines 593-600 (conclusion): “In conclusion, the current study provides new insight into the temporal dynamics and development of face-related expectations. Notably, our findings raise the notion that the formation of cued expectations does not occur at random within the period preceding a statistically expected target. Instead, the facilitation of this developmental process appears to be instigated by the predictive image and proceeds, with fluctuations in growth, until shortly before the depiction of the target. In turn, expected stimuli are met by a relatively precise expectation to allow the brain to reserve cognitive resources. The evolution of implicit face-related expectations, thus, seems to prevail over the entire interstimulus period.”

#3 In lines 76-78, authors write that "The insight that the projection of prediction errors is mediated by high gamma frequency (60 - 100Hz) synchronisation, which requires a greater energetic cost than lower frequencies [8,9]". This statement is understandable but how does that shine light on the underlying processes shaping variability of ERPs is not clear. Perhaps, you could consider revising this statement.

We very much appreciate that the reviewer drew our attention to the fact that the sentence mentioned above does not clearly state how varying the amount of allocated processing resources shapes ERPs. We have revised the entire paragraph to make this statement more precise and comprehensible. (Lines 75-96)

Lines 75-96: “To establish such predictions, the brain draws upon prior information of expected events in preparation for their actual occurrence [8]. Recent studies have shown that pre-activation of sensory information, and subsequent sensory priors, are mediated by low frequency oscillations encompassing alpha and beta frequency ranges [8-12]. These oscillations, primarily alpha, are believed to enhance the signal-to-noise ratio in task-related networks by carefully selecting relevant and simultaneously silencing irrelevant populations of neurons to establish a more focused access to representations of expected stimuli [13]. Due to the early access to relatively precise prior information, less cognitive resources are required to process anticipated perceptual input and in turn visual event-related potentials are modulated [7,9,13]. Additionally, updating and optimising this given neural representation would be unnecessary, hence, the forward projection of prediction errors is downregulated. Bottom-up processing as well as the projection of prediction errors functionally relate to high gamma frequency (60 – 100Hz) synchronisation, which requires a greater energetic cost than lower frequencies [14,15]. Based on these findings, a reduction in gamma power would be presumed to proceed the onset of expected events. In contrast, due to the limited access to pre-activated prior information, more cognitive resources would be allocated to processing unexpected occurrences [10,15,16]. Unexpected events could therefore be distinguishable from expected occurrences by enhanced post-stimulus GBA, whereas expected images are preceded by an enhancement in pre-stimulus alpha/beta power and a suppression in GBA post-stimulus onset [11]. In turn, whilst less cognitive resources are devoted to processing sensory information of expected targets, subsequently evoking a diminished ERP response, the opposite would be expected for a novel or surprising occurrence [7,9].”

#4 Also, please explain why lesser Cognitive resource would suggest diminished ERPs. Is there already worked out inverse relationship between no. of Cognitive Resource and amplitude/power of ERP response or vice versa. Could please cite appropriate reference to justify this relationship?

We thank the reviewer for pointing out that the description of the relationship between the allocation of cognitive resources and modulations of ERPs is vague and in need of supporting references. As mentioned above (point 3), we have revised the corresponding paragraph to clarify this relationship. (Lines 75-96)

#5 The opposite such as amplified ERPs are evident for arbitrary or surprising occurrences. This is of course similar to what you expect for familiar versus unfamiliar stimuli or novel stimuli which is not consolidated yet in the memory. However, those which are already consolidated via prior experience/learning are already very familiar (e.g., expected faces) and should give similar to repetition suppression effect (repeated exposure). Similarly, for surprising faces one should expect very similar effect as oddball stimuli. I would appreciate if you could you provide some of your reasoning against this comment?

We thank the reviewer for this interesting comment. Our main reason against the notion that the N170 amplitude for random images was amplified due to surprise or novelty, is based on the observed modulation in pre-stimulus alpha/beta power. This enhancement in alpha/beta power prior to expected images (versus random images) suggests early prioritisation of top-down processes. Especially the positive correlation between the last peak in alpha/beta power and the modulation of the N170 proposes a relationship between these top-down processes and early visual processing of faces. Instead of an amplitude amplification for the random images, as reflected by oddball stimuli, the obtained results in this study seem to be more coherent with the notion that the N170 differences between random and expected images were elicited by expectation.

#6 Authors write "Drawing upon this, it seems apparent that somewhat unexpected events are predominantly distinguishable from expected occurrences by enhanced gamma-band activity (GBA)." There could be multiple interpretation of enhanced gamma-band activity. One specific reason you mention is supported by few recent studies which suggest that enhanced GBA could be involved in encoding and updating internal representation. However, thinking loud along the line of reasoning you have outlined in the Introduction I wonder whether this also has anything to do with deployment of larger number of cognitive resources while processing somewhat unexpected events. If you could kindly comment on this.

We thank the reviewer for this valuable point. We agree with the reviewer that the observed enhancement in GBA for random images (relative to expected images) could be due to several reasons, such as an increase in allocated cognitive resources. We have revised the corresponding section in the Introduction (see point 3). Furthermore, to link the discussion of GBA back to the revised reasonings outlined in the Introduction, we have also addressed this notion more precisely in the Discussion.

Lines 483-488: “Acknowledging that the occurrence of the face images without a preceding predictive image lacked the predictability of the paired images, random images were deemed to require more cognitive resources and elicit an enhanced gamma-band response. In other words, since all images were task relevant and required a specific behavioural response, it seems likely that more cognitive resources were necessary for processing randomly occurring images, for which the gender was not foretold by a predictive image.”

Materials and methods

#7 You do not have a gender balanced sample for this study. I am wondering whether that has any impact on the gradual built up of implicit expectation and top-down prediction of expected face occurrences. Previous studies have shown that in the primary visual cortex while decoding faces and expressions, women showed a more bilateral functioning than men and there are some gender differences while processing neutral and affective faces. There are also difference that are found in the EEG N1 response as well as during oddball N170 amplitude response. Could you please provide your reasoning?

We thank the reviewer for drawing our attention to this literature. Although a gender balanced sampled would have been preferred, it was unfortunately not an option as there was a lack of male participants who signed-up for this study. However, the focus of the present study was to investigate the general origination and development of perceptual expectations, irrespective of gender differences and/or individual differences. Regardless, as rightfully pointed out by the reviewer, previous studies have shown sex-related differences in lateralisation of the N170 when processing faces. Interestingly, Proverbio and colleagues (2012) showed that instead of the previously observed bilateral functioning in women and a more right-lateralised dominance in men, sex-coding studies showed a different pattern in hemispheric lateralisation [Proverbio AM., et al. (2012). Neuropsychologia]. Here, women showed a more dominating left-lateralised response, whilst men showed bilateral functioning; thus, suggesting that the involvement of the left hemisphere is essential during gender classification in both gender groups. Given that the ‘cover-up’ task in our study was a gender classification task, the observed left-lateralised modulation of the N170 would be concurrent with the results obtained by Proverbio et al. (2012). To take this reasoning into account, we have extended the discussion regarding the left-lateralised modulation of the N170 (lines 463-476). 

Nevertheless, the unbalanced sample size (10 men and 23 women) makes it difficult to draw firm conclusions regarding any sex-related differences impacting hemispheric lateralisation. Based on the current participant sample, we cannot fully dismiss that the gradual build-up of expectations may or may not differ between gender groups. However, this intriguing question is beyond the scope of the present study and would benefit from an adequately designed study that specifically investigates how sex-related difference may impact the origination of face-related expectations. 

Lines 463-476: “On a different note, past studies have shown sex-related difference in face processing and the lateralisation of the N170. These findings suggest a dominating right lateralisation of the N170 in men and a more bilateral tendency in women [36,37]. Intriguingly, Proverbio and colleagues (2012) showed that sex-coding studies revealed a slightly different pattern in hemispheric lateralisation [38]. Here, women showed a more dominating left-lateralised response whilst men showed bilateral functioning; thus, suggesting that the involvement of the left hemisphere is essential during gender classification in both gender groups. Given that in the present study participants performed a gender classification task, the observed left-lateralised modulation of the N170 may have been influenced by the underlying nature of the task at hand and the fact that women outnumbered men (10 men and 23 women). However, this unbalanced sample makes it difficult to draw firm conclusions regarding any sex-related differences impacting hemispheric lateralisation. Ultimately, this question would be interesting to pursue in future, with an adequately designed study that specifically investigates how sex-related difference may impact the origination of face-related expectations.”

#8 The authors hypotheses regarding the enhancement Alpha/Beta power pre-stimulus as top down prediction of expected faces. I am wondering why hypothesize about amplitude/power and why not phase of pre stimulus Alpha/Beta. I would imagine phase of pre stimulus Alpha may also play a crucial role. Also, this change in pre stimulus Alpha power could just reflect the ongoing background excitability which has a biophysical basis and has got nothing to do with any prediction about the faces. Please comment on this.

We thank the reviewer for these two valuable comments. In the current study, our main aim was to locate a timepoint within the pre-stimulus timeframe at which the origination of a face-related expectation commenced, and how this build-up would develop and change over time. No specific timepoints and frequencies were therefore isolated a priori. Instead, the entire pre-stimulus timeframe and all frequencies within alpha and beta ranges were considered. Since time-frequency (power) representations allow for spectral and temporal ‘source’ separation over a vast number of frequencies and timepoints, we believed a spectral power analysis to be most suitable for our approach. In future studies, it would, however, be very interesting to examine alpha/beta phase coherence at the timepoints of each of the observed peaks. We would therefore like to thank the reviewer for suggesting this investigative idea for follow-up studies. We have subsequently added a sentence to the Discussion which addresses phase coherence (lines 590-592).

Line 590-592: “In addition, although phase analysis is beyond the scope of this current study, it would be an intriguing research question for future studies to investigate alpha/beta phase coherence at the timepoints of each of the observed peaks.”

We have addressed the reviewer’s thought-provoking comment on background excitability below. Please see comment #14. 

#9 In lines 214-215, authors have mentioned about the selection of pre-post durations, however, not provided any details with regards to what is the basis of this selection.

The authors write, "-200ms pre- to 600ms post-stimulus onset for ERPs. The 200ms prior to stimulus onset served as a baseline. For the TFA, data was epoched from -2000ms to approx. 1500ms, time-locked to stimulus onset."

Any comment on what exactly guided your selection of a specific pre stimulus duration as we know from previous studies that the choice of pre stimulus duration can have significant impact on the outcome of power changes and involvement of specific frequency band of interest. Hence, what frequency bands subserve the prediction process is very much sensitive to the choice of those parameters e.g., pre stimulus duration.

We thank the reviewer for pointing out that the reasons guiding our choice in epoch sizes were not mentioned in Methods and materials. For the N170, the recommended epoch length beneficial for the analysis of “early” ERPs was used [Cohen MX. (2014). Analyzing Neural Time Series Data (pp. 73-85); Luck S. (2014). An Introduction to the Event-Related Potential Technique (pp. 71-117)]. In terms of time-frequency, we were primarily interested in the timeframe prior to stimulus onset for alpha/beta frequencies (entire pre-stimulus timeframe), as well as post-/peri-stimulus onset for gamma frequencies (0 – 1000ms). We therefore decided on a segment that framed stimulus onset. The choice of a relatively long timeframe of 3500ms was motivated by the intention to allow edge artefacts (usually lasting 2-3 cycles) to subside before and after our points of interest [Cohen MX. (2014). Analyzing Neural Time Series Data (pp. 73-85)]. Hence, since 250ms on either side of the epoch were removed through convolution, at least 2 cycles of each of our frequencies of interest (8 – 30Hz) were removed prior to subsequent analyses. We have modified the corresponding section in the manuscript accordingly (lines 234-236). 

Lines 234-236: “These time segments of 3500ms framing image onset were used with the intention to allow edge artefacts to subside before and after our points of interest [22].” 

Results

#10 Kindly explain why a title as "Early alpha/beta oscillations reflect the gradual formation of face-related expectations in the brain" deemed appropriate.

Is it really gradual over the entire period of pre stimulus duration or is it an accelerated process just before the onset of the expected image? It seems to be the latter based on the following results unless I did not follow these results correctly.

A significant positive relationship was, however, observed between the modulation of alpha/beta power underlying the final peak (8-30Hz; 2500 - 3000ms) and the modulation of the N170 (Spearman's rho = .46, p = .021, 95% 381 CI [0.14 0.69]; Fig 5).

We agree with the reviewer, that we can only conclusively say that the final peak showed a positive relationship between alpha/beta power and the reduction of the N170. Nevertheless, substantial differences in alpha/beta power appeared to prevail from the onset of the predictive image until the occurrence of the expected image (relative to random images). Past accounts have proposed that an increase in alpha/beta power reflects inhibition of forward feeding networks to impede information processing [Klimesch W. (2011). Brain Research; Klimesch W., Sauseng P., Hanslmayr S. (2007). Brain Research Reviews]. Arguably, it seems very plausible that a similar neural state is elicited upon the presentation of the predictive image. Meaning, the continuous modulation in alpha/beta power could, therefore, provide a prolonged favourable condition in which relevant top-down processes are prioritised whilst irrelevant, competing representations are suppressed. Especially since each predictive image only cued a single specific face, alternative neural representations were unnecessary to be processed or maintained during this interval. Thus, even though only the last peak correlated with the modulation of the N170, the onset of the predictive image seems to instigate an early optimisation in neural state to boost the development of relatively precise perceptual expectations. In our eyes, it is therefore hard to completely disregard this facilitatory activity. The term ‘gradual’ was therefore used with the intention to encompass all underlying facilitatory activity within the interstimulus timeframe that is likely to aid the evolution of cued perceptual expectations. However, we can see that this choice in wording might cause confusion and have subsequently removed it from the title. 

Discussion

#11 I enjoyed reading the discussion section. However, I could not resist myself repeating my last comment again. Discussion section actually begins with what was my last comment for the previous section. It seems to me a more appropriate title hiding in the statement below,

"Predictable visual events are often met by implicit expectations to allow the brain to reserve cognitive resources." Authors should certainly clarify which one of the two is more appropriate interpretation here the gradual development and built up of predictions prior to expected stimulus onset or an accelerated process showing tendency towards conserving cognitive resources in the process of predictions of expected outcome.

We thank the reviewer for their positive feedback concerning the Discussion. Furthermore, we are thankful that the reviewer draws our attention to the fact, that our choice in wording may be misleading. We have revised the corresponding sections in the manuscript to emphasise that this continuous enhancement in alpha/beta power could reflect a favourable neural state that optimises early prioritisation of top-down processes in order to facilitate the development of implicitly cued expectations.

Lines 402-405: “Collectively, these findings suggest that the final peak could reflect a relatively precise expectation of the upcoming stimulus. The continuous enhancement of alpha/beta power extending throughout the entire interstimulus interval, may, on the other hand, provide an elongated favourable state optimal for expectation formation.” 

Lines 412-421 (start of Discussion): “The present study provides findings to suggest that the development of implicitly cued expectations is optimised by the early prioritisation of top-down processes. In turn, predictable visual events are met by relatively accurate implicit expectations to allow the brain to reserve cognitive resources. These processes were reflected by enhancements in pre-stimulus alpha/beta power for expected relative to randomly occurring faces. Intriguingly, this enhancement commenced as early as the onset of the predictive image and prevailed until the expected stimulus occurred. A correlation between the final elevation in alpha/beta power and the reduction of the N170 revealed a positive relationship between these two modulations. Ultimately, a reduction in bottom-up processing for expected relative to random images was reflected by a suppression in post-stimulus gamma power (Fig 6).”

Lines 520-522 (middle of Discussion): “It appears that the initial activation of underlying processes facilitating expectation formation is subsequently triggered by the informative attribute of these cue-like images.”

Lines 529-543 (middle of Discussion): “This continuous modulation in alpha/beta power thus seems to suggest that the prioritisation of top-down processes commences much earlier than just immediately prior to the occurrence of the expected target. Several past accounts have provided evidence to suggest that alpha/beta power is an electrophysiological marker for the inhibition of forward feeding networks [13,42]. Arguably, it seems very plausible that a similar neural state is elicited upon the presentation of the predictive image. As such, the predictive image seems to give rise to a favourable condition in which increases in alpha/beta power reflect prioritisation of top-down processes whilst competing forward-feeding representations are suppressed. Especially since each predictive image only cued a single specific face, alternative neural representations were unnecessary to be processed or maintained during this interval. The reverse has been demonstrated recently in a study by Griffith et al., (2019), which showed that a decrease in alpha/beta power (disinhibition of relevant networks) facilitates information processing [42]. Thus, the continuous maintenance of a favourable condition within the timeframe confined by the onsets of the predictive and expected images could appear to aid the development of precise perceptual expectation.”

Lines 569-572 (end of Discussion): “Even though the early peaks did not correlate with the modulation of the N170, the onset of the predictive image seems to initiate an early optimisation of a favourable neural state to boost the development of relatively precise perceptual expectations.”

#12 In lines 426-427 I suggest please provide some relevant references when you are discussing right lateralized habituation effects.

"These habituation effects or memory-driven modulations were found to be predominantly right-lateralised." (reference please)

 Is this memory-driven modulations of N170 ERPs predominantly right-lateralised irrespective of gender in face related expectation. Related to my earlier question about selected sampling.

Foremost, we thank the reviewer for drawing our attention to the missing references. The corresponding references have been added to line 455. In addition, we are glad that the reviewer raises the intriguing question, that the right-lateralised, memory-driven modulations of the N170 could be gender driven. We have surveyed the referenced literature and found that two of the studies collected data from a sample with a dominance in female participants (12 female and 8 male participants [Caharel S., et al. (2009). Neuropsychologia]; 8 female and 5 male participants [Jemel B., Schuller AM., Goffaux V. (2010). J Cogn Neurosci]). On the other hand, the third study was conducted with only male participants [Campanella A., et al. (2000). Psychophysiology]. Together, these results suggest that the memory-driven modulation of the N170 seems to be primarily right-lateralised regardless of the participants’ gender. We have inserted this observation into line 455.

Lines 455-456: “These habituation effects or memory-driven modulations were found to be predominantly right-lateralised, irrespective of gender [32,33,34].”

#13 I think the authors cited and covered references which are most relevant to their findings based on ERPs, pre stimulus brain activity, GBA.

However, I would like to point out that the recent studies suggest frequently reported negative relationship between pre-stimulus α/Beta power and stimulus detection or behavioural performance may be explained by changes in detection criterion. I am wondering whether there is something similar is happing here for example Pre stimulus power change in Alpha/Beta is driven by the state of neural excitability (brain states of individual participants), rather than top-down prediction of expected occurrences. Could you please comment on this. Related to my earlier question on background excitability.

We thank the reviewer for this thought-provoking comment. Given our experimental paradigm, we agree that it would seem likely that modulations in pre-stimulus alpha/beta power could have been driven by neural excitability. That is, a shift in detection criterion, regulated by for instance confidence, regarding the upcoming presentation of an expected versus a somewhat unexpected image. However, our neurophysiological results do not quite coincide with those of previous studies. Samaha and colleagues (2017) showed that increased neural excitability (decrease in pre-stimulus alpha) correlated with increased confidence ratings attributed to a given stimulus-related response [Samaha J., Iemi L., Postle BR. (2017). Conscious Cogn]. In terms of perceptual sensitivity, decreased pre-stimulus alpha power has been linked to an increase in the detection of stimuli [Limbach K., Corballis PM. (2016). Psychophysiology]. Both studies mentioned show that the topological distribution of these modulations in pre-stimulus alpha power are located over posterior electrodes, which is a well-replicated finding in studies involving visual stimulation. In both cases, the decrease in alpha power extended from 500ms pre-stimulus until stimulus onset. On the contrary, we observed a more bilateral alpha/beta power distribution, ranging over temporal and parietal electrodes within the 500ms timeframe prior to stimulus onset (Fig 4Bii). Furthermore, our results suggest that an enhancement in alpha/beta power precedes expected relative to random images (Fig 4B: expected - random images). Namely, a reduced pre-stimulus alpha/power for random images (versus expected). When compared to the results reported in the mentioned studies above, it would seem questionable why participants would show heightened confidence regarding the upcoming approach of a random image compared to the predictable onset of an expected image. Collectively, it seems more likely that the observed enhancement in pre-stimulus alpha/beta power indicates top-down predictions for forthcoming expected occurrences rather than increases in neural excitability. 

#14 The authors write in line 512 One could question why this pre-stimulus enhancement in alpha/beta power fluctuates instead of being a stable and continual increase in power leading right up to the presentation of expected targets.

I do have the same question and also related to my previous question of gradual bult up of expectation versus final peak increase before the presentation of expected targets. You have already provided an elaborate and solid reasoning in lines 515-521. But still curious whether this transient enhancement of power in the Alpha/Beta band should be considered at all for the entire inter-stimulus duration of 3 seconds as you have mentioned is too long a duration. Somehow, it seems these individual peaks are fundamentally functionally distinct and more meaningful pre stimulus anticipatory power change is occurring over a relatively short-interval of time frames with regards to significant functional attenuation the amplitude of ERPs to the presentation of expected targets. Hence, the clear role of these three peaks at least the first two and in particular, the middle one is not so clear after all. The waxing and waning of pre stimulus Amplitude/power change over certain time intervals and windows could be purely based on the change in background excitability and brain states.

I could also anticipate substantial inter individual and inter-trial differences in the processes of how expectations reflected in increased alpha/beta activity, which makes processing of expected stimuli more efficient for some participants and consequently reduces the amplitude of the N170 but not necessarily with the same efficiency and reduction for others.

We thank the reviewer for these thoughtful points. With reference to points 10 and 11, we have revised the Discussion to highlight and clarify that the first two peaks could indeed reflect optimisations in brain states, which in turn prioritise top-down processing and facilitate the development of cued expectations. Furthermore, we agree with the reviewer that the efficiency with which the expected images are processed will probably vary between trials and participants. We believe this to be a crucial point that should be addressed in future single-trial (phase coherence) studies.

---

## [Decision Letter · Decision Letter 1]

12 Jul 2021

Early alpha/beta oscillations reflect the formation of face-related expectations in the brain

PONE-D-21-12218R1

Dear Dr. Roehe,

We’re pleased to inform you that your manuscript has been judged scientifically suitable for publication and will be formally accepted for publication once it meets all outstanding technical requirements.

Kind regards,

Mukesh Dhamala, Ph. D.

Academic Editor

PLOS ONE

Additional Editor Comments (optional):

We accept the work for publication, but ask the authors to consider doing a minor revision of the manuscript following the comment of Reviewer 1.

Reviewers' comments:

Reviewer's Responses to Questions

**Comments to the Author**

1. If the authors have adequately addressed your comments raised in a previous round of review and you feel that this manuscript is now acceptable for publication, you may indicate that here to bypass the “Comments to the Author” section, enter your conflict of interest statement in the “Confidential to Editor” section, and submit your "Accept" recommendation.

Reviewer #1: All comments have been addressed

Reviewer #2: All comments have been addressed

2. Is the manuscript technically sound, and do the data support the conclusions?

Reviewer #1: Yes

Reviewer #2: Yes

3. Has the statistical analysis been performed appropriately and rigorously? 

Reviewer #1: Yes

Reviewer #2: Yes

4. Have the authors made all data underlying the findings in their manuscript fully available?

Reviewer #1: Yes

Reviewer #2: Yes

5. Is the manuscript presented in an intelligible fashion and written in standard English?

Reviewer #1: Yes

Reviewer #2: Yes

6. Review Comments to the Author

Reviewer #1: I appreciate that the authors have carefully undertaken the revisions following my previous comments. I have a minor issue related to point 6 of the first round review.

In Abstract there is a statement “A particularly interesting finding was the early onset of alpha/beta power enhancement which peaked immediately after the depiction of the predictive face; meaning, three seconds prior to the presentation of an expected image. Hence, our findings propose an approximate timeframe throughout which consistent traces of enhanced alpha/beta power illustrate the early prioritisation of top-down processes to facilitate the development of implicitly cued face-related expectations.”

This statement needs to toned down/ rephrased because the 3 second time-scale is resulting from a choice of a specific stimulus and not true for any face related expectations presented in an arbitrary time scale.

Reviewer #2: (No Response)

7. PLOS authors have the option to publish the peer review history of their article (what does this mean?). If published, this will include your full peer review and any attached files.

Reviewer #1: **Yes: **Arpan Banerjee

Reviewer #2: No

---

## [Editor Report · Acceptance letter]

16 Jul 2021

PONE-D-21-12218R1 

Early alpha/beta oscillations reflect the formation of face-related expectations in the brain 

Dear Dr. Roehe:

I'm pleased to inform you that your manuscript has been deemed suitable for publication in PLOS ONE. Congratulations! Your manuscript is now with our production department. 

Kind regards, 

on behalf of

Dr. Mukesh Dhamala 

Academic Editor

PLOS ONE